



# On the multi-fractal scaling properties of sea ice deformation

**Pierre Rampal[1], Véronique Dansereau[1], Einar Olason[1], Sylvain Bouillon[3],
Timothy Williams[1], and Abdoulaye Samaké[2]**

[1]Nansen Environmental and Remote Sensing Centre and Bjerknes Centre for Climate Research,
Bergen, Norway
[2]Université de Bamako, Bamako, Mali
[3]Numéca, Louvain La Neuve, Belgium

Correspondence to: Pierre Rampal (pierre.rampal@nersc.no)



## Abstract

In this paper, we evaluate the neXtSIM sea ice model with respect to the observed scaling invariance properties of sea ice deformation in the spatial and temporal domains. Using an Arctic set-up with realistic initial conditions, state-of-the-art atmospheric reanalysis forcing and geostrophic currents retrieved from satellite data, we show that the model is able to reproduce the observed properties of these scaling in both the spatial and temporal domains over a wide range of scales and, for the first time, their multi-fractality. The variability of these properties during the winter season are also captured by the model. We also show that the simulated scaling exhibit a space-time coupling, a suggested property of brittle deformation at geophysical scales. The ability to reproduce the multi-fractality of these scaling is crucial in the context of downscaling model simulation outputs to infer sea ice variables at the sub-grid scale, and also has implication in modeling the statistical properties of deformation-related quantities such as lead fractions, and heat and salt fluxes.

## 1 Introduction

The fact that sea ice deformation maps look similar at different scales, with highly localized deformation features intersecting with a wide range of intersection angles (Hutchings et al., 2005; Wang, 2007, e.g.,), suggests scale-invariance in the spatial domain (Erlingsson, 1988). We note that scale-invariance in space is also observed in sea ice for other deformation-related quantities, such floe sizes (Rothrock and Thorndike, 1984; Matsushita,1985) and keel profiles (Rothrock and Thorndike, 1980). Comprehensive datasets of sea ice drift are now available at different spatial and temporal resolutions, from 50 m/10 min (Oikkonen et al., 2017), 400 m/2 days (Thomas et al., 2004, 2007, 2009) to 5-10 km/3 days (Kwok, 2001; Stern and Moritz, 2002). Analyses of these datasets have confirmed the presence of scale-invariance and, in particular, has confirmed that sea ice deformation is highly localized in both space and time.





In the spatial domain, deformation is observed to be concentrated along quasi one-dimensional, so-called kinematic linear features (LKFs) organized in "web-like arrays" (Kwok et al., 1998; Thomas et al., 2007) that can be clearly identified over a wide range of space scales (Thomas et al., 2007; Linow and Dierking, 2017). Sea ice deformation also appears to be a self-similar, highly localized process in the time domain. Isolated, short-duration deformation events of various levels of intensity occur over a wide range of frequencies. These events also sustain deformation, maintaining the LKFs "active" for many days Coon et al. (2007). The reorganization and formation of new LKFs occur in response to changes in the large scale atmospheric forcing (Kwok, 2001) and permanent deformation with high deformation rates in the ice pack is mainly synchronous with high winds events (Oikkonen et al., 2017).

A quantitative indication of *scale-invariance* in sea ice deformation is given by the shape of the distribution of deformation rate invariants, such as the shear, divergence and total deformation rates, which we refer to here as $\dot{\varepsilon}$. These probability density functions ($P$) have indeed been shown to be "heavy-tailed", i.e., dominated by extreme values that are out of the Gaussian basin of attraction, following a power law decay of the type

$$P(\dot{\varepsilon}) \sim \dot{\varepsilon}^{-\gamma}, \tag{1}$$

where $\gamma$ is an exponent larger than 1 that depends on the spatial and time scale considered (Lindsay and Stern, 2003; Marsan et al., 2004; Rampal et al., 2008; Hutchings et al., 2011; Bouillon and Rampal, 2015b). This important characteristic expresses scale-invariance, as it is impossible from a power law distribution to determine the scale of a given deformation even by comparing the relative number of deformation events of different sizes.

*Localization* in the time and space domain is revealed by scaling analysis of the deformation rate invariants. In such analysis, deformation rates are estimated at different spatial and temporal scales, by such methods as "coarse-graining". The *mean* deformation rate, estimated by coarse-graining analysis (e.g., Lindsay and Stern, 2003; Marsan et al., 2004; Bouillon and Rampal, 2015b; Rampal et al., 2016), or pair of buoys dispersion analysis (Rampal et al., 2008), have been shown to vary with the spatial scale, $L$, and temporal





scale of observation, $T$, as

$$\langle\dot\varepsilon\rangle \sim L^{-\beta}, \tag{2}$$
$$\langle\dot\varepsilon\rangle \sim T^{-\alpha}, \tag{3}$$

respectively, hence following a power law. The exponents, $\beta$ and $\alpha$ are both equal or greater
than zero and quantify the degree of localization of the deformation. In the space domain,
$\beta = 0$ characterizes the homogeneous deformation of an elastic solid or viscous fluid, i.e., a
deformation that does not depend on the spatial scale, while $\beta = 2$ corresponds to a single
linear fracture concentrating all of the deformation in an otherwise undeformed material.
Conversely, in the time domain, $\alpha = 0$ corresponds to a homogeneous deformation and a
single, temporally isolated deformation event corresponds to the limit of $\alpha = 1$. This scaling
has been shown to hold over a very wide range of space and time scales (Rampal et al.,
2008; Oikkonen et al., 2017; Weiss, 2017), with that $\alpha$ and $\beta$ larger than 0, even for time
scales on the order of the winter season and for space scales on the order of the length
of the Arctic basin. This result indicates the absence of a characteristic time and/or space
scale for the mean sea ice deformation over these scales and, as a consequence, that sea
ice deformation can not be approximated homogeneous over time/space scales relevant for
the Arctic Ocean.

The fact that sea ice deformation is characterized by extreme events out of the Gaussian
basin of attraction also indicates that the mean (moment of order 1) is not a sufficient
quantity to describe fully the distribution of deformation rates at a given time/space scale
and therefore not a sufficient basis for temporal and scaling analyses. Higher moments of
the distribution of deformation rates, such as the variance (order 2) and skewness (order
3), should therefore also be explored to describe the entire distribution and the associated
process of sea ice deformation.

While the value of $\beta$ for the first moment (the mean) describes the rate at which the
magnitude of deformation events decreases with the scale of observation, it is the change
in the value of $\beta$ and $\alpha$ with respect to the moment $q$ of the distribution that indeed indicates
how the temporal and spatial localization itself changes with the *magnitude* of deformation





events. This change can be described by structure functions of the form

$$\beta(q) = aq^2 + bq, \tag{4}$$
$$\alpha(q) = cq^2 + dq, \tag{5}$$

in space and time respectively. In the case of a linear structure function, i.e., no curvature, the amount of localization of large and small deformation events is the same and the scaling is said to be *mono-fractal*. For both coefficients $a$ and $b$ or $c$ and $d > 0$, the functions are quadratic and convex. The higher order moments of the distribution increase much faster than the lower order moments with decreasing scale of observation. This indicates that *large* deformation events are *more localized* in time and space than smaller events. This effect is stronger as the curvature is higher and in itself is the definition of multifractal *heterogeneity* and *intermittency*.

Spatial scaling analysis of sea ice deformation retrieved from radar or buoy drift data have shown a clear multi-fractal scaling expressed by a power law scaling of the first, second, and third moments, ranging from the resolution of the data up to hundreds of km (e.g., Marsan et al., 2004; Rampal et al., 2008; Hutchings et al., 2011; Bouillon and Rampal, 2015a). Recently, Weiss and Dansereau (2017) have suggested, based on the combination of all available data, including the ones of Oikkonen et al. (2017), that this multi-fractality also holds in the time domain, over a period of 3 to 160 days. We note that multi-fractality in space has also been observed for open water densities (Weiss and Marsan, 2004) and lead fractions (Olason et al., 2019), and in time for shear stress amplitudes (Weiss and Marsan, 2004) and principal stress directions (Weiss, 2008).

These properties of sea ice deformation imply that observations of these quantities available at large scales can be statistically related, i.e, downscaled, to the same quantities at smaller, unresolved scales. In the case of model simulations, *downscaling* of outputs could be particularly valuable to infer quantities at the sub-grid and/or sub-time-step scale. In this context, the capability to reproduce mono- versus the multi-fractality of these properties becomes very important. Indeed, if one was to estimate the distribution of a variable at the sub-grid scale based on a model that would not reproduce the observed multi-fractality,





but only a mono-fractality, then the downscaled distribution of this variable would greatly underestimate extreme values.

Despite the large number of interesting studies on the subject and the numerous hypothesis of its significance for sea ice rheology (e.g., Weiss and Dansereau, 2017), no numerical model was yet shown to be able to reproduce the multi-fractal behaviour of sea ice deformation in *both* the space and time domains.

The observed self-similarity and multi-fractality in the deformation and related characteristics of sea ice actually poses great challenges to the development of sea ice models, in particular in the continuum framework. On the one hand, the momentum and evolution equations for sea ice properties are based on *mean* variables. On the other hand, however, the observed multi-fractality in sea ice deformation implies that there is not a clear separation of scales between the strain rate due to mesoscale (50-100 m) heterogeneities in the ice (leads, ridges, etc.) and the strain rates at 10 to 100 km scale. Consequently, no scale appears appropriate to homogenize sea ice motion and thereby define a mean velocity or deformation rate for model resolution ranging from 50 m to 100 km.

In the absence of a characteristic space/time scale for the sea ice deformation and with the knowledge that localization goes beyond the space/time resolution of typical regional and global models, perhaps the best a continuum framework for sea ice modelling can do is to localize the deformation at the smallest available or *nominal* scale. This is one of the major objective in developing neXtSIM, the numerical model used in the current study. Models with high localization capabilities are otherwise essential in the view of allowing an accurate representation of sea ice-ocean-atmosphere interactions, in both the contexts of short-term and climate predictions.

This paper consists in the last step in validating neXtSIM against sea ice deformation statistics. While previous work have shown that the model reproduces the observed scaling of sea ice deformation (Bouillon and Rampal, 2015a; Rampal et al., 2016) in *space*, the temporal scaling and multi-fractality of both types of scaling have not yet been demonstrated. The comparison performed here is based on satellite observations of sea ice deformation and winter-long simulations over the Arctic Ocean.





The first part of the paper discuss the recent developments of neXtSIM, the simulation setup and the observations (Section 2). The second part describes the methodology used to perform the multi-fractal scaling analyses on both the model and observational data (Section 3). Results of these analyses are presented in Section 4 and discussed in Section 5.

## 2 Model and observations

### 2.1 Model and simulation setup

neXtSIM is a finite elements sea ice model that uses a moving Lagrangian mesh. Its original dynamical component was based on the Elasto-Brittle (EB) mechanical framework of (Amitrano et al., 1999), first implemented in the context of sea ice by Girard et al. (2011) to account for brittle fracturing processes and the associated spatial localization of deformation. This framework was later adapted by Bouillon and Rampal (2015b) and Rampal et al. (2016) for long-term simulations of the Arctic sea ice cover including thermodynamical processes and advection, using a Lagrangian treatment of the equations of motion and a dynamical remeshing scheme. Year-long simulations were presented in Rampal et al. (2016) and evaluated with respect to sea ice area, extent, volume, drift, and deformation. In particular, the simulated deformation rates were demonstrated to be in good agreement with observations on the basis of their scaling properties *in space*.

However, the Elasto-Brittle model does not, by definition, include a physical mechanism for *irreversible* deformations, as it is based on a strictly linear-elastic constitutive law. It therefore cannot represent the transition between the small, elastic deformations associated with the fracturing of the ice cover and the permanent, potentially large, post-fracturing deformation that dissipates internal stresses. It is therefore not suited to represent the dynamical behavior of a fractured ice cover over long (>day) time scales and cannot represent fully the properties of sea ice deformation *in time*.



The recent Maxwell-Elasto-Brittle (MEB) rheology addresses this limitation of the EB framework by including a mechanism for the relaxation of internal stresses that depends on the degree of fracturing of the sea ice cover (Dansereau et al., 2016). It is implemented in the current version of neXtSIM, which is used for this study. Another addition to the model is the introduction of a three-thickness-categories scheme that represents explicitly the thin and newly-formed ice. The other model components (thermodynamics, slab ocean, etc.) remain unchanged relative to the version presented by Rampal et al. (2016).

All of the relevant equations entering the current version of neXtSIM are presented in the Appendix (Sections A1 for the dynamical core and A2 for the three-thickness-categories scheme and sea ice thermodynamics). The numerics (spatial and temporal discretizations, advection scheme and numerical solvers) are the same as described in Rampal et al. (2016).

The initial mesh is generated in pre-processing over a pan-Arctic region by using the mesh generator presented in Remacle and Lambrechts (2016) with a prescribed mean resolution (length of the vertices of the triangular elements) of 10 km. The coasts are defined from the Global Self-consistent, Hierarchical, High-resolution Geography Database[1]. The domain is restricted to the central Arctic by putting open boundaries on the lines cutting the Bering Strait from (-166.0 , 67.7 N) to (-170.7, 65.7 N) and cutting the Canadian Arctic Archipelago from (-59.0, 76.7 N) to (-121.0 ,69.5 N) and on the 2-segments line cutting the Greenland and Barents and Kara Seas by joining the coordinates (-19.0, 77.0N), (11.0, 73.0 N), (22.0, 72.9 N), (43.9, 76.1 N), (75.4, 75.7 N) and (88.5, 73.6 N). We checked that using a larger domain with open conditions much further from the zone of interest does not impact the results presented in this paper.

The atmospheric forcing consists of the applied 10 m wind velocity, the 2 m air temperature, the mixing ratio, the mean sea level pressure, the total precipitation amount and snow fraction, and the incoming short-wave and long-wave radiation. All of these quantities are

---

[1]GSHHS_f_L1.shp, downloaded from https://www.ngdc.noaa.gov/mgg/shorelines/data/gshhg/latest/gshhg-shp-2.3.5-1.zip, accessed 1 February 2017





provided as three-hourly means from the atmospheric state of the Arctic System Reanalysis[2] (Bromwich et al., 2016).

The ice-ocean surface stress is computed from monthly ocean surface geostrophic currents derived as in Armitage et al. (2017) from the Arctic sea surface height data obtained from altimeters by Armitage et al. (2016). The provided fields have a hole of missing data around the North Pole that we filled using a linear interpolation between the northernmost available points and their mean. A smoother is applied to the ocean velocity components in the filled area to avoid spurious oscillations due to the interpolation method. The slab ocean salinity and temperature are nudged towards the daily sea surface temperature and salinity data from the TOPAZ4 reanalysis[3] (Sakov et al., 2012) with a nudging time scale equal to 30 days. TOPAZ4 is a coupled ocean and sea ice data assimilation system for the North Atlantic and the Arctic that is based on the HYCOM ocean model and the ensemble Kalman filter data assimilation method using 100 dynamical members. A 23-year reanalysis has been completed for the period 1991–2013 and is the multi-year physical product in the Copernicus Marine Environment Monitoring Service. The ocean depth, $H$, used for the basal stress parametrization comes from the 1 arc-minute ETOPO1 global topography[4] (Amante and Eakins, 2009).

Our reference simulation starts on November 15th, 2006. The level of damage of the ice cover (see Appendix A1) is initially set to zero where sea ice is present. Initial sea ice concentration and thickness are set from a combination of the TOPAZ4 reanalysis, and the OSISAF climate data record (Tonboe et al., 2016) and ICESAT[5] Kwok et al. (2009) datasets respectively.

---

[2]https://rda.ucar.edu/datasets/ds631.0, ASRv1 30-km, formerly ASR final version, Byrd Polar Research Centre/The Ohio State University. Accessed 15 April 2015

[3]available at http://marine.copernicus.eu/services-portfolio/access-to-products/

[4]available at https://www.ngdc.noaa.gov/mgg/global/

[5]available at https://icdc.cen.uni-hamburg.de/1/daten/cryosphere/seaicethickness-satobs-arc.html





## 2.2 Satellite observations

We use the Lagrangian displacement data produced by the RADARSAT Geophysical Processor System (RGPS) as described in Kwok et al. (1998). This dataset covers the Western Arctic for the period 1996–2008 and provides trajectories of sea ice "points" initially located on a 10 km regular grid (http://rkwok.jpl.nasa.gov/radarsat/lagrangian.html). The positions of these points are updated when two successive SAR images are available. The time interval between two updates is typically 3 days. For the present analysis we use the data covering the winter period 2006-12-03 to 2007-04-30.

## 3 Methodologies for scaling analysis

Scaling analyses of sea ice deformation can be performed with two approaches: the coarse-graining method as in Marsan et al. (2004) and buoy dispersion analysis as in Rampal et al. (2008). We use the coarse-graining approach in this study. It is applied on velocity gradients computed at the resolution of the trajectory dataset, using triplets of points. The nominal resolution is defined as the square root of the surface area of the polygon considered. For example, the resolution that can be achieved with the RGPS dataset is about 7.5 km when using 3-sided polygons obtained from Delaunay triangulation.

Drifters in the model are seeded at the location of the RGPS grid points as of December 3, 2016. The RGPS grid for this initialization is undeformed and the points are regularly spaced by 10 km. The positions of the simulated drifters are updated at each model time step until the end of the simulation or until the ice concentration drops to zero (through melting or opening of a lead). Both the RGPS and simulated trajectories are filtered for the presence of coasts, with a proximity threshold of 100 km. Only the trajectories that are common to both the simulation and RGPS dataset are considered in the calculation of the deformation and their statistics.

Triplets of drifting points are defined as the result of Delaunay triangulation of the initial positions of the tracked RGPS points, which ensures that the associated polygons are in-





dependent, i.e., non-overlapping. The polygons after initiation are defined by the positions of their three nodes at any given time. We stress the fact that the simulated trajectories are not reinitialized every 3 days to match the RGPS positions; only the initial positions are identical between the RGPS and the model trajectories.

5 Coarse-graining in space is obtained by performing Delaunay triangulations on the sub-sampled cloud of initial RGPS drifter positions. Each set of polygons obtained using this process will be associated to a spatial scale, $L$, defined as the mean of the square root of the polygon surface areas. The number of triplets available for the statistical analyses decreases as the space scale increases. Coarse-graining in time is performed by considering 10 the positions of triplets of drifters separated by a time $T$. The number of available triplets also decreases as the time scale increases.

For each available polygon, the total deformation rate is calculated as:

$$\dot{\varepsilon}_{tot} = \sqrt{\dot{\varepsilon}^2_{shear} + \dot{\varepsilon}^2_{div}} \tag{6}$$

where $\dot{\varepsilon}_{shear}$ and $\dot{\varepsilon}_{div}$ are the two invariants, shear and divergence respectively, of the 15 deformation rate. These invariants are estimated using a contour integral calculation as follows: The spatial derivatives of the components $u$ and $v$ of the velocity calculated at a given time scale $T$ are obtained by calculating the contour integrals as in Kwok et al. (2008) and Bouillon and Rampal (2015b) around the boundary of each polygon associated to a given space scale $L$:

$$u_x = \frac{1}{A} \oint u\,dy \tag{7}$$

$$u_y = -\frac{1}{A} \oint u\,dx \tag{8}$$

$$v_x = \frac{1}{A} \oint v\,dy \tag{9}$$

$$v_y = -\frac{1}{A} \oint v\,dx, \tag{10}$$





where $A$ is the encompassed area of the polygon equal to $L^2$. For example, $u_x$ is approximated by:

$$u_x = \frac{1}{A} \sum_{i=1}^{n} \frac{1}{2}(u_{i+1} + u_i)(y_{i+1} - y_i), \tag{11}$$

where $n = 3$ and subscript $n + 1 = 1$. The shear rates $\dot{\varepsilon}_{shear}$ and divergence rates $\dot{\varepsilon}_{div}$ are then computed as:

$$\dot{\varepsilon}_{shear} = \sqrt{(u_x - v_y)^2 + (u_y + v_x)^2}, \tag{12}$$

$$\dot{\varepsilon}_{div} = u_x + v_y. \tag{13}$$

The distribution of total deformation rates is constructed from each given coupled space/time scale $(L, T)$, and their first 3 moments are calculated as $\langle \dot{\varepsilon}_{tot}^q \rangle$ where $q = 1, 2, 3$ is the moment order.

Below we discuss some issues that are inherent to the data and coarse-graining method and their impact in terms of the robustness of the statistics calculated here.

– With time, the triangular elements can become too distorted, in which case their length scale, $L$, is poorly defined. Applying a test for distortion based on the smallest angle of the polygons and discarding the most distorted ones was found to affect the results in terms of the *slope* of the scaling, and the goodness of the fit of the power law fit of the scaling. Hence here we discard from the analysis the polygons having a minimum angle of 30 degrees.

– The RGPS trajectories are not sampled at regular time intervals, as the model is, due to the irregular interval between two satellite orbits. The mean sampling is of about 3 days, and 90% of trajectories are sampled with a frequency between 2.5 and 3 days. We found that using different sampling times for the observations and the model affect the comparison results. To deal with this issue, we performed a sub-sampling



of the RGPS trajectory dataset using a nearest-neighbour interpolation of the original positions at 3-days intervals. The positions simulated by the model are taken to match the sub-sampled RGPS time series.

– The 3-days RGPS sampling additionally places a lower bound on the time scales one can explore when comparing the simulated and observed deformation rates. In the present analysis, we therefore chose to not explore smaller time scales.

We find on the whole that the relative number of available polygons is what has the largest impact on the deformation statistics. Some facts therefore need to be kept in mind when performing a scaling analysis over a *finite* period of time. In the time domain, in particular, this entails that sea ice deformation is better sampled, i.e., more triplets are available, for the early than for the late part of the period. In the present case, the computed statistics are therefore more representative of early than late winter. This effect is even more important for the larger time scales: polygons separated by small time scales $T$ will indeed approximately sample the entire period while for large time scales, more polygons will be available at the beginning than at the end of the period.

## 4  Results

Figure 2 shows the maps of the 3-days shear deformation rates simulated by the model and estimated from the RGPS data at the same locations and for the same period of 7 days centered on 4 February 2007. The cumulative probability of the simulated and observed shear deformation rates from the snapshots of Figure 2 are shown on Figure 1. Both distributions exhibit a power-law tail, with almost identical slopes of about -3, similar to what e.g., Marsan et al. (2004) found in their study. This implies that one needs to consider higher moments than the mean and standard deviation of the distributions to fully describe the statistics of the sea ice deformation process (Sornette, 2006). In the scaling analysis presented in the following sections, we thus systematically calculate the 3 first moments of the distributions of deformation rates.





## 4.1 Spatial scaling analysis

Figure 3 (left panel) shows the winter mean of the spatial scaling analysis for the observations and model. We found that both model and observations statistics are following power-laws. As suggested in Stern et al. (2018), we use logarithmically spaced bins and applied an ordinary least squared method to the binned data in log-log space to get reasonably accurate estimate of the power-law fits. The mean deformation rates are very well captured by the model at all scales. The second and third moments of the distributions are, however, slightly underestimated by the model compared to the observations for scales lower than about 40 km. For example, at the nominal scale of 7.5 km, the second and third moments are underestimated by a factor of 2 and 3 respectively compared to the observations. This may be due to one or several of the following factors: (1) inaccuracies in the atmospheric forcing (2) our choice of mechanical parameters values and (3) the value of the atmospheric drag coefficient. It is especially important to note that the simulated deformation rate has not been tuned with respect to every mechanical parameters in the present simulations. We consider such tuning to be out of the scope of this study, which focuses on the ability to reproduce the observed scaling and, in particular, their multi-fractal property. The simulated and observed structure functions $\beta(q)$ are, however, equal within their margin of error (Figure 3, right panel). The error bars are estimated from the minimal and maximal local scaling exponent values, as in Bouillon and Rampal (2015a) and correspond to *upper-bound* estimates.

In both cases, the scaling is clearly multi-fractal, as no linear function can be contained within the error bars. Instead, both structure functions are obtained by applying a quadratic fit to the data (in the least squared sense) as defined by equation (4). The good agreement between the observed and modelled structure functions is a relevant indication that the scaling in the simulated deformation is consistent with that observed between 7.5 and 580 km.

A time-series of the value of the scaling exponent for the mean obtained for the successive and contiguous *snapshots* throughout the winter is shown on Figure 4 (left panel). It

(c) Author(s) 2019. CC BY 4.0 License.



shows that the scaling exponent varies between -0.1 and -0.34. These exponents are in good agreement with the 1-month running means of the scaling exponents calculated by Stern and Lindsay (2009) for the entire period covered by the RGPS dataset (1996-2008). The scaling exponent for the mean is about 0.2 on average over the whole winter period for the simulated and observed total deformation rates, which is the value found by Marsan et al. (2004) for a snapshot of deformation rates calculated over a 3-days period centred on 6 November 1997. We note also that the model reproduces well the observed variability of the scaling exponent throughout the whole period. A time-series of the value of the curvature (parameter $a$ in equation (4)) is also calculated for that period (Figure 4, right panel). It shows that the curvature values fluctuate within the range 0.03-0.13. The value of the curvature, corresponding to the level of multi-fractality of the scaling and indicating the degree of heterogeneity of the deformation fields, is about 0.07 on average for the model and about 0.08 for the observations over the winter period analyzed here.

We further characterize the properties of the spatial scaling for both the model and observations by exploring its dependence on the temporal scale, $T$. We find that the estimated spatial scaling exponent, $\beta$, decreases with increasing $T$ (Figures 5 and 6, left panels). This is the signature of the space-time coupling of the scaling properties of sea ice deformation, originally suggested in Rampal et al. (2008) and further explained in Marsan and Weiss (2010) as being a characteristic of brittle deformation at geophysical scales. This property is for the first time shown from a sea ice model simulation. The origin of this coupling has been previously proposed to be linked to the complex correlation patterns related to chain triggering of ice-quakes. Further study is, however, needed to explore this hypothesis, which is out of the scope of this paper.

We also note a decrease of the multi-fractal character of the spatial scaling when increasing the time scales from $T = 3$ to $T = 96$ days (Figures 5 and 6, right panels). For the model, we transition from a multi-fractal to a mono-fractal scaling, while for the observations the scaling remains clearly multi-fractal at all temporal scales considered in this study. The curvature values are decreasing from 0.085 to nearly zero for the model and from 0.16 to 0.09 for the observations following a power-law (Figure 7). The general behaviour of de-



creasing the degree of multi-fractality of the spatial scaling as the time scale increases is thus captured by the model, but the model fails at keeping the multi-fractal signature at the largest scales. This may come from the fact that the highest deformation events are too evenly distributed over the Arctic region in the simulation compared to the observations. The reason for this discrepancy should be further explored but is out of scope of the present paper.

## 4.2 Temporal scaling analysis

The results of the temporal scaling analysis for $L = 7.5$ km is shown on Figure 8 (left panel). We see a robust and very similar power-law scaling for both the model and observations between $T = 3$ days (i.e., the temporal resolution of the observations) and $T = 96$ days. In previous studies based on drifting buoy trajectories whose positions are sampled at higher frequency, it has been suggested that the temporal scaling for the mean total deformation holds down to 1 hour (Hutchings et al., 2011). A recent study based on very high resolution ship radar measurements has demonstrated that it holds down even to 10 min (Oikkonen et al., 2017). Here, we obtain a perfect agreement between the slope (about -0.3) of the temporal scaling for the mean deformation rates estimated in this recent study, and that estimated from the RGPS data and the present model simulations (gray, dark and cyan top curves in the left panel of Fig. 8).

We note, however, that the third moment of the distributions are slightly underestimated by the model at all time scales. This means that the proportion of extreme deformation events compared to lower ones is too small or that their values are too low in the simulation. This may come from the inaccuracy of the relatively coarse (30 km) atmospheric reanalysis we use to force our model and that is known to poorly resolve the most *extreme* low pressure systems, a common shortcoming of all the available global or regional atmosphere reanalysis to date. Another explanation could be the fact that we have not tuned the MEB rheology parameters to reproduce the proportion of extreme deformation events versus the lower ones. In this rheology, the coupling between the damage and the mechanical behavior of sea ice is non-linear and it is therefore expected that varying parameter



values can change the proportion of the simulated extreme events, i.e., the skewness of the distribution of deformation rates.

As in the spatial domain, the temporal scaling is found to be multi-fractal for the model and observations, and the match is virtually perfect. The quadratic functions $\alpha(q)$ gives curvature values of 0.11 for the model and 0.13 for the observations, the exact same value as the one found by Weiss and Dansereau (2017) (figure 1), despite the fact that they analyzed a different period (winter 1996-1997). This seems to argue that the multi-fractality of the temporal scaling is a robust property of sea ice deformation, at least in the winter time, independent of the observed change in sea ice cover state and the associated shift of its dynamical regime during the period 1996-2006 (e.g., Rampal et al., 2009a, b).

We also investigate the dependence of the temporal scaling on the spatial scale of observation, $L$, for both the model and RGPS data (Figures 9 and 10, left panels). We find that the scaling exponent, $\alpha$, decreases with $L$. Similar to the spatial scaling analysis performed in Section 4.1, we find here the signature of a space-time coupling in the scaling properties of sea ice deformation. The multi-fractal character of the temporal scaling holds at all the spatial scales considered here ($L = 7.5$ to $L = 360$ km), and is similar in the model and observations (Figures 9 and 10, right panels). The curvature values are going from 0.11 down to 0.015 for the model and from 0.13 to 0.01 for the observations following a power-law (Figure 11). The decrease in the degree of multi-fractality of the temporal scaling as the space scale increases as seen in the observations is remarkably well captured by the model.

## 5 Discussion

Our statistical analyses have shown that the neXtSIM model reproduces correctly the distribution of sea ice deformation rates, its scaling properties in both the space and time domains and its multi-fractal behavior. In particular, it is the first time that multi-fractality in the time domain is shown to be reproduced in a sea ice model.



The MEB rheology was developed with the aim of improving the representation of the physics of sea ice continuum models by including the ingredients hypothesized by Weiss and Dansereau (2017) to be the cause of the emergence of multi-fractal heterogeneity and intermittency of sea ice deformation. This hypothesis is based on the analysis of observa-
5 tional data that have highlighted the multi-fractality of sea ice deformation in space and time (Rampal et al., 2008; Bouillon and Rampal, 2015b; Weiss and Dansereau, 2017) as well as on the close analogy with other systems such as the Earth crust as proposed originally in Weiss et al. (2009). According to Weiss and Dansereau (2017) the ingredients required are: a *threshold mechanism* for brittle fracturing, some *disorder* that represents the nat-
10 ural heterogeneity of the material at the mesoscale, *long-range elastic interactions* within the ice cover that promote avalanches of fracturing events through a cascading mecha- nism, post-fracturing relaxation of elastic stresses through *viscous-like relaxation*, and a *slow restoring/healing mechanism* of the sea ice mechanical properties. We argue that the results obtained here are an important step towards the confirmation of this hypothesis.
15 We show here that the spatial scaling of sea ice deformation simulated in a realistic setup by neXtSIM holds down to the nominal resolution of the mesh, a result that is in agreement with previous analyses of the MEB model in idealized simulations (Dansereau et al., 2016) and realistic ones (Rampal et al., 2016). It means that neXtSIM does not need to be run at higher spatial resolution in order to resolve the presence of linear kinematic features (e.g.,
20 running at about 1 km resolution in order to resolve sea ice deformation at scale of about 10 km). We show also that this spatial localization and the multi-fractal character of the simulated mean sea ice deformation is resolution-independent in this setup. This is what is shown on figure 12. However, and despite the fact that the scaling remains multi-fractal when neXtSIM runs at coarser resolutions (e.g., 15 or 30 km), the level of multi-fractality is
25 decreasing with decreasing resolution. Indeed, the second and third moments of deforma- tion rates from the 15 and 30 km runs differ from the results obtained from the 7.5 km run (figure 12, right panel), which suggests an underestimation of extreme deformation events at the smaller spatial scales with increasing model resolution. Nevertheless, the represen- tation of multifractality at all resolutions implies that neXtSIM could be adequately used to





explore a wider range of space-time scales than that covered by the currently available observations of the global Arctic. In particular, it could allow to "zoom in" and explore the statistical properties of sea ice deformation at the sub-satellite observations scales, which are of increasing interests for both regional to global climate modelling and operational forecasting. A model that could otherwise not represent the multi-fractal character of sea ice deformation and would only reproduce a mono-fractal scaling would greatly underestimate extreme deformation events and their impact on sea ice conditions at such scales like e.g., the presence or not of leads and ridges.

A model that allows reproducing sea ice deformation and its scaling properties down to its nominal resolution does not preclude the need for appropriate sub-grid scale parametrizations. On the contrary, we believe that physically sound parametrizations are indeed required and that the knowledge of the distribution of deformation rates at the the sub-grid scale made possible by neXtSIM could be highly valuable in terms of informing these parametrizations. An appropriate sub-grid scale parametrization links the deformation simulated at the scale of the grid cell with the scale at which deformation really occurs within the ice cover, which is the size of individual leads and ridges.

We moreover argue that, as sea ice deformation is strongly tied to other model variables, such as drift, lead fraction and thickness distribution. A proper simulation of these variables is a necessary prerequisite to using models for investigating various coupled ocean–ice–atmosphere processes, and their impact on their immediate vicinity and on the polar climate system. For example, the accuracy of neXtSIM in reproducing the observed statistical properties of sea ice deformation as demonstrated in this paper is thought to go hand-in-hand with its capability in representing the observed properties of lead fraction. This is the subject of a concurrent study presented in Olason et al. (2019).



## 6  Conclusions

In this study we have compared the deformation rates simulated by neXtSIM to those derived from observations by comparing their distributions and how these distributions scale in time and space. The conclusions of our analysis are:

- The neXtSIM model reproduces well the first, second and third moments of the statistical distribution of observed sea ice deformation rates and how it scales in space and time.

- Sea ice deformation rates calculated over a temporal scale of 3 days scale in space from the scale of the model/observations up to about 700 km in a multi-fractal manner.

- Sea ice deformation rates calculated over a spatial scale of 7.5 km scale in time over the range 3 days–3 months in a multi-fractal manner.

- A space-time coupling in the scaling properties of sea ice deformation is for the first time shown to be reproduced by a model. This suggests that neXtSIM could be a proper tool to study the physical meaning and origin of this coupling, in the context of brittle deformation of geophysical solids.

- The simulated mean sea ice deformation rates and their associated scaling invariance characteristics are resolution-independent, i.e., when running the neXtSIM model at resolutions of 7.5, 15 or 30 km. The most extreme deformation events may be missed however if running at coarser resolutions, i.e. the second and third order moments may be underestimated compared to the high-resolution run.

- As the mono versus multi-fractal character of the scaling of deformation rates is the discriminating factor for the heterogeneity and intermittency of the deformation, we suggest that a multi-fractal scaling analysis should be a prerequisite validation step before further analyzing sea ice model outputs that could be influenced by sea ice dynamics.




– The good agreement between the model and observations motivates the use of neXtSIM as a tool to further investigate physical processes that are highly sensitive to sea ice deformation.

## Appendix A: Model description

This section presents the dynamical and thermodynamical components of neXtSIM. The wave-in-ice module implemented by Williams et al. (2017) is not included here. Prognostic sea ice variables are listed in Table 1 and all parameter values are listed in table 2.

### A1 Dynamical core

The evolution equation for sea ice velocity comes from vertically integrating the horizontal sea ice momentum equation as follows:

$$\rho_i H \frac{D\boldsymbol{u}}{Dt} = \nabla \cdot \left(H\boldsymbol{\sigma}\right) + \boldsymbol{\tau}_a + \boldsymbol{\tau}_w + \boldsymbol{\tau}_b - \rho_i H \left(f\boldsymbol{k} \times \boldsymbol{u} + g\nabla\eta\right). \tag{A1}$$

The parameter $\rho_i$ is the ice density, $H$ is the mean ice thickness per unit grid cell area, $\boldsymbol{\tau}_a$, $\boldsymbol{\tau}_w$ and $\boldsymbol{\tau}_b$ are the surface wind, ocean and basal stresses, respectively, and are defined as in Rampal et al. (2016). The parameter $f$ is the Coriolis frequency, $\boldsymbol{k}$ is the upward pointing unit vector, $g$ is the acceleration due to gravity and $\eta$ is the ocean surface elevation. In the region with only thin ice or with thick ice thickness lower than a given threshold (defining our ice edge), the momentum equation is replaced by a Laplacian equation so that the velocity linearly decreases from the ice edge to the nearest coast (see Samaké et al. (2017)). The additional ice pressure term introduced in Rampal et al. (2016) is not included here.

Following Dansereau et al. (2016), the evolution equation for the internal stress takes the form of the Maxwell constitutive law:

$$\frac{D\boldsymbol{\sigma}}{Dt} + \frac{\boldsymbol{\sigma}}{\lambda} = E\boldsymbol{K} : \dot{\varepsilon}(\boldsymbol{u}) \tag{A2}$$





where $\lambda$ is the relaxation time for the stress, $E$, is the elastic modulus and $\dot{\varepsilon}$, the strain rate tensor, is defined as the rate of strain tensor

$$\dot{\varepsilon}(\boldsymbol{u}) = \frac{1}{2}\Big(\nabla\boldsymbol{u} + (\nabla\boldsymbol{u})^T\Big). \tag{A3}$$

Plane stresses conditions are assumed and the stiffness tensor $\boldsymbol{K}$ reads

$$\begin{bmatrix} (\boldsymbol{K}:\varepsilon)_{11} \\ (\boldsymbol{K}:\varepsilon)_{22} \\ (\boldsymbol{K}:\varepsilon)_{12} \end{bmatrix} = \frac{1}{1-\nu^2} \begin{pmatrix} 1 & \nu & 0 \\ \nu & 1 & 0 \\ 0 & 0 & \frac{1-\nu}{2} \end{pmatrix} \begin{bmatrix} \varepsilon_{11} \\ \varepsilon_{22} \\ 2\varepsilon_{12} \end{bmatrix} \tag{A4}$$

where $\nu$ is Poisson's ratio.

As in Dansereau et al. (2016), both the elastic modulus, $E$, and the relaxation time are functions of the ice concentration, $A$, and the level of damage, $d$. The level of damage is a scalar, grid-scale variable that represents the density of fractures at the sub-grid scale. Its value is 0 for an undamaged and 1 for a "completely" damaged material. The elastic modulus is a linear function of $d$:

$$E(A,d) = E_0(1-d)f(A), \tag{A5}$$

where $E_0$ is the undamaged elastic modulus and $f(A)$ introduces a dependence on the ice concentration via the following exponential function:

$$f(A) = e^{c^*(1-A)}, \tag{A6}$$

where $c^*$ is the ice compactness parameter introduced by Hibler (1979). As in Dansereau et al. (2016), the relaxation time is a power function of $d$:

$$\lambda(d) = \lambda_0(1-d)^{\alpha-1}, \tag{A7}$$

where $\lambda_0$ is its undamaged value and $\alpha$ is a constant exponent greater than 1. Here, we use the values $\alpha = 4$ and $\lambda_0 = 10^7 s$ ( 115 days) (as in the realistic Maxwell-EB simulations





of Dansereau et al., 2017) to ensure that the relaxation of stresses is virtually zero over an undamaged ice cover but is significant when the ice is damaged.

The evolution of the damage is controlled by the location of the predicted stress state relative to the failure envelope, which as in Rampal et al. (2016) is defined in terms of the principal stress components

$$\sigma_1 = -\frac{\sigma_{11} + \sigma_{22}}{2} + \sqrt{\left(\frac{\sigma_{11} - \sigma_{22}}{2}\right)^2 + \sigma_{12}^2} \tag{A8}$$

$$\sigma_2 = -\frac{\sigma_{11} + \sigma_{22}}{2} - \sqrt{\left(\frac{\sigma_{11} - \sigma_{22}}{2}\right)^2 + \sigma_{12}^2}, \tag{A9}$$

with the convention that compressive stresses are positive.

Here, the envelope combines a Mohr-Coulomb failure criterion and a maximum tensile and compressive stress. The three criteria are given by

$$\sigma_1 - q\sigma_2 \leq \sigma_c g(H) \quad \text{(Mohr-Coulomb criterion)}, \tag{A10}$$

$$-\frac{\sigma_1 + \sigma_2}{2} \leq \sigma_{T\max} g(H) \quad \text{(tensile stress criterion)}, \tag{A11}$$

$$\frac{\sigma_1 + \sigma_2}{2} \leq \sigma_{N\max} g(H) \quad \text{(compressive stress criterion)}, \tag{A12}$$

where $q = \left[\left(\mu^2 + 1\right)^{1/2} + \mu\right]^2$, $\sigma_c = \frac{2c}{\left[\left(\mu^2 + 1\right)^{1/2} - \mu\right]}$, $\mu$ is the internal friction coefficient, $c$ is the cohesion, $\sigma_{T\max}$ is the maximal tensile strength and $\sigma_{N\max}$ the maximum compressive strength (see table 2). The cohesion, $c$, is scaled as a function of the model spatial resolution, as described in Bouillon and Rampal (2015a).

When one of the damage criteria is met, $d$ is modified according to

$$1 - d' = \Psi(1 - d), \tag{A13}$$



where

$$
\Psi = \begin{cases} \dfrac{\sigma'_c}{\sigma_1 - q\sigma_2} & \text{if } \sigma_1 - q\sigma_2 > \sigma'_c \\ \dfrac{2\sigma'_{T\,\text{max}}}{-\sigma_1 + \sigma_2} & \text{if } -\dfrac{\sigma_1 + \sigma_2}{2} > \sigma'_{T\,\text{max}} \\ \dfrac{2\sigma'_{N\,\text{max}}}{\sigma_1 + \sigma_2} & \text{if } \dfrac{\sigma_1 + \sigma_2}{2} > \sigma'_{N\,\text{max}}. \end{cases}
\tag{A14}
$$

Healing is included here to represent the counteracting effect of refreezing of water within leads on the level of damage of the ice cover. It is implemented via a constant term in the damage evolution equation:

$$
\frac{Dd}{Dt} = \frac{(1-d)(1-\Psi)}{T_d} - \frac{1}{T_h},
\tag{A15}
$$

where $T_h$ is the characteristic time for healing and $T_d$, the characteristic time for damaging (Dansereau et al., 2016).

## A2 Ice thickness redistribution and thermodynamics

neXtSIM includes the 3 ice categories suggested by Stern and Rothrock (1995): thick ice, thin ice and open water. In our implementation the thin ice is only newly formed ice, so ice will only be transferred from the thin-ice category to thick ice, but not in the reverse direction. Thin ice is described by its concentration, $A_t$, and volume per unit area, $H_t$. Thick ice is described by the concentration, $A$, and volume per area, $H$. We assume that the thin ice has no mechanical strength and simply follows the motion of the surrounding thick ice.

Thin ice thickness is considered to be uniformly distributed between $h_{min} = 5$ cm and $h_{max} = 50$ cm so that the volume per unit area is bounded between $H_{min} = Ah_{min}$ and $H_{max} = A\frac{h_{min} + h_{max}}{2}$. The evolution equations for $A$, $H$, $A_t$ and $H_t$ have the following form:

$$
\frac{D\phi}{Dt} = -\phi\nabla \cdot \boldsymbol{u} + \Psi_\phi + S_\phi,
\tag{A16}
$$





where $\dfrac{D\phi}{Dt}$ is the material derivative that is defined for any scalar and vector as

$$\frac{D\phi}{Dt} = \frac{\partial\phi}{\partial t} + (\boldsymbol{u}\cdot\nabla)\phi. \tag{A17}$$

Here $\nabla\cdot\boldsymbol{u}$ is the divergence of the horizontal velocity, $\Psi_\phi$ a sink/source term due to ridging, and $S_\phi$ a thermodynamical sink/source term. Volume conservation is imposed by setting $\Psi_H = -\Psi_{Ht}$ and an additional constraint is that $A_h + A \leq 1$.

The evolution of $A$, $H$, $A_t$ and $H_t$ is computed following three main steps:

1. Advection: The scheme solves the equation:

$$\frac{D\phi}{Dt} = -\phi\nabla\cdot\boldsymbol{u}, \tag{A18}$$

for each conserved scalar quantity ($A$, $H$, $A_t$, $H_t$, etc.). For this paper, we use the purely Lagrangian scheme presented in Rampal et al. (2016). After this step the concentration could be larger than 1.

2. Mechanical redistribution: The scheme imposes the limit $A_t + A \leq 1$ on the total ice area by following those steps:

   (a) Compute the new open water concentration as:

   $$A_0 = \max(0, 1 - A - A_t), \tag{A19}$$

   a source term for the open water could be added here (as in Stern and Rothrock, 1995) to represent sub-grid scale convergence/divergence.

   (b) Compute the new thin ice concentration as:

   $$A_t^{n+1} = \max(0, \min(1, 1 - A - A_0)) \tag{A20}$$



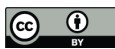

(c) Compute the transfer of thin ice if $A_t^{n+1} < A_t$ by setting:

$$H_t^{n+1} = H_t \frac{A_t^{n+1}}{A_t} \tag{A21}$$

$$H^{n+1} = H + \left(H_t - H_t^{n+1}\right) \tag{A22}$$

$$\Delta A = \frac{A_t - A_t^{n+1}}{\zeta} \tag{A23}$$

where $\zeta$ is an aspect ratio parameter set to 10.

(d) Compute the new thick ice concentration as:

$$A^{n+1} = \max(0, \min(1, 1 - A_t^{n+1} - A_0 + \Delta A)) \tag{A24}$$

(e) Apply more ridging if $(A^{n+1} + A_t^{n+1}) > 1$ by setting $A^{n+1} = 1 - A_t^{n+1}$

3. Growth/melt: The source/sink terms from the thermodynamics are computed by applying the zero-layer Semtner (1976) vertical thermodynamics to the new ice category and that of Winton (2000) for the thick ice, as if the thickness was uniform and equal to $H/A$ for the thick ice and $H_t/A_t$ for the thin ice. Freezing of open water is computed as in Rampal et al. (2016) such that heat loss from the ocean that would cause super cooling is redirected to ice formation. The newly formed ice is transferred to the thin ice category and is assumed to have a thickness equal to $h_{min}$. The transfer from the thin ice to the thick ice and the lateral melting of thin ice is computed by applying the bounding limit $H_{min}$ and $H_{max}$.

*Author contributions.* This work is the result of a long-term team effort at the Nansen Centre in Norway carried by Sylvain Bouillon, Einar Olason, Pierre Rampal, Abdoulaye Samaké, Timothy Williams to develop the new sea ice model `neXtSIM` and include the Maxwell-Elasto-Brittle (MEB) rheology of Dansereau et al. (2016). PR led the research and performed the analyses presented in this paper; VD and PR led the writing with contributions from EO, SB and TW; AS implemented the parallel C++ version of the model code described in Samaké et al. (2017), with support from SB and EO; SB and VD implemented the MEB rheology.





*Acknowledgements.* This manuscript was prepared thanks to the financial support of the Research Council of Norway (RCN) through the *FRASIL* project (grant no. 263044). However, the development of the neXtSIM model has been supported since 2013 through several projects from the RCN, in particular the *SIMECH* (grant no. 231179), and the *NEXTWIM* (grant no. 244001) projects. TOTAL E&P is also thanked for their continuous support over the period 2013-2017 through the *KARA* project. And finally, we thank Jerome Weiss and David Marsan for the fruitful discussions.

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





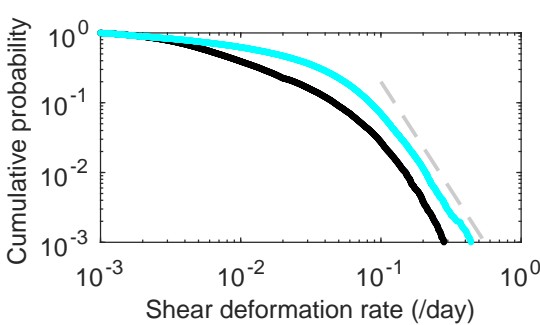

**Figure 1.** Cumulative probability functions of the shear deformation rates shown on figure 2 for the model (cyan) and the observations (black). The deformations are calculated over a time scale of 3 days, and a spatial scale of 7.5 km (mean of the squared root of triangle's surface areas and for which the deformations are calculated). The dashed line is for reference and corresponds to a power-law with an exponent equal to -3.



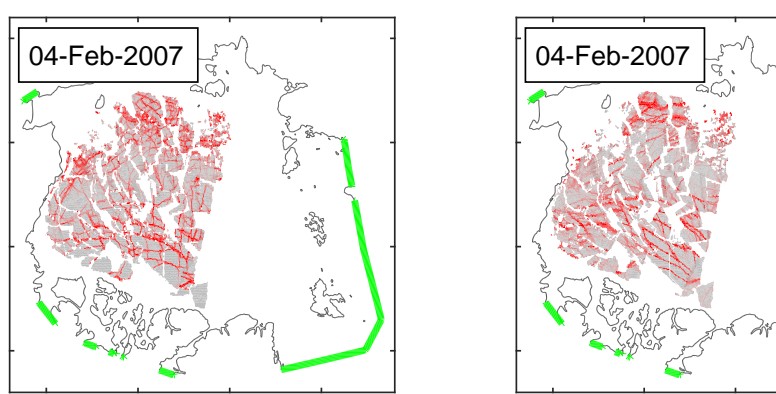

**Figure 2.** Shear sea ice deformation rate in per day, as simulated by the model (left) and observed from satellite (right). The deformation is calculated over a time scale of 3 days, for the period of 7 days centred on 4 February 2007





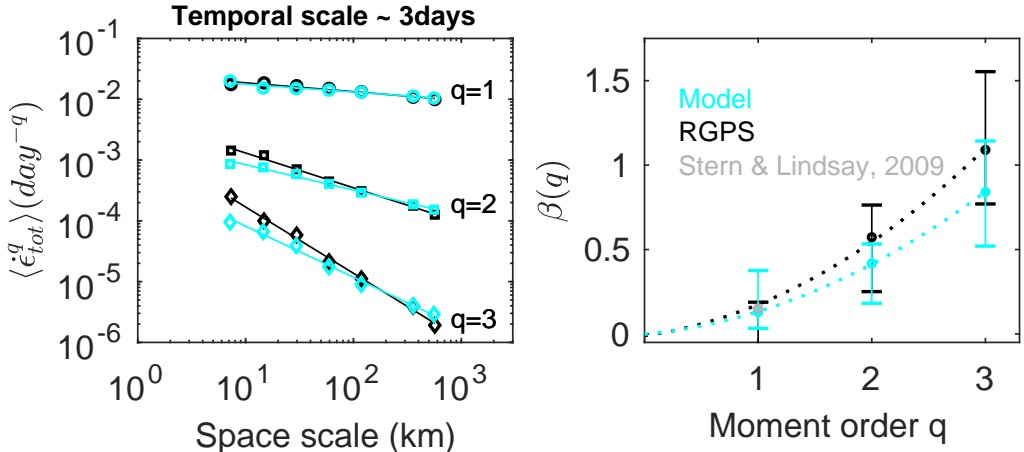

**Figure 3.** Spatial scaling analysis of the observed (black) and simulated (blue) total deformation rate calculated over a time scale of 3 days from the motion of the same triplets in the model than in the RGPS dataset. Left panel: Power law fits $\langle \dot{\varepsilon}^q \rangle \sim L^{-\beta(q)}$ for the moments $q = 1, 2$ and 3 of the distributions of the shear deformation rate $\dot{\varepsilon}$ calculated at different spatial scales $L$ are shown as solid lines. Right panel: Corresponding structure functions $\beta(q)$ for both the model and observation, where $\beta$ indicates the exponent of the power laws fits, and $q$ is the moment order are shown as dashed lines. The error bars are estimated from the minimal and maximal local scaling exponent as in Bouillon and Rampal (2015a) and thus correspond to upper-bound estimates.





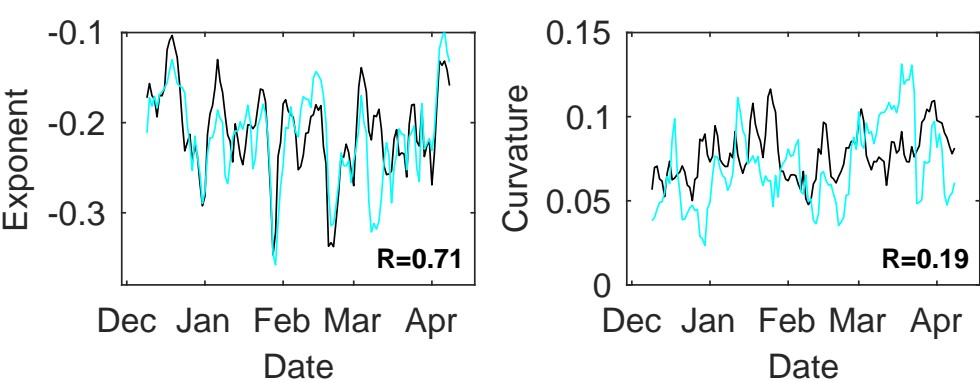

**Figure 4.** Time series of the power scaling exponents (left) and of the curvature of the structure function (right) for the model (cyan) and the observations (black).



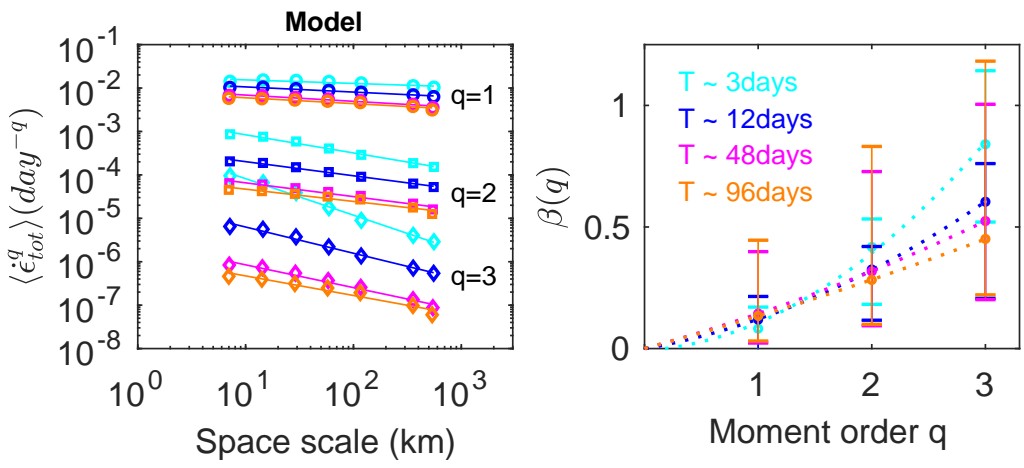

**Figure 5.** Same as Figure 3 but for the model at various temporal scales.





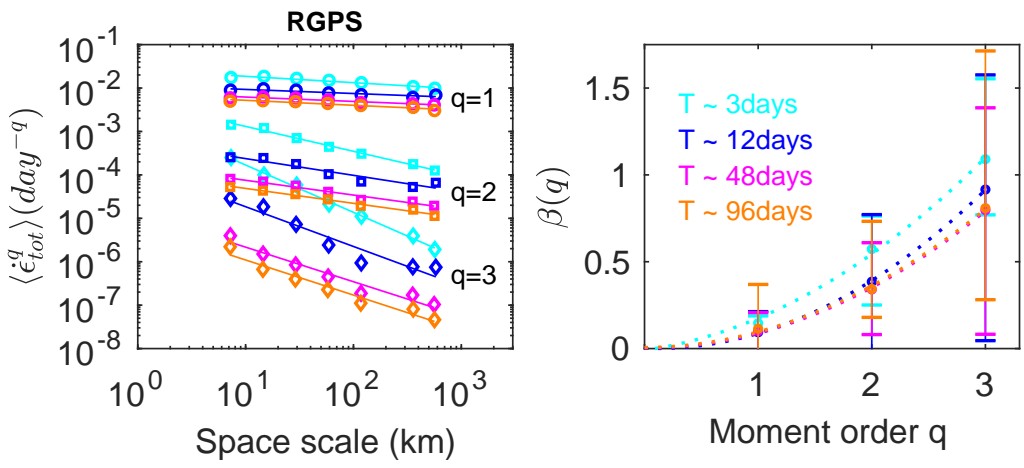

**Figure 6.** Same as Figure 3 but for the observations at various temporal scales.



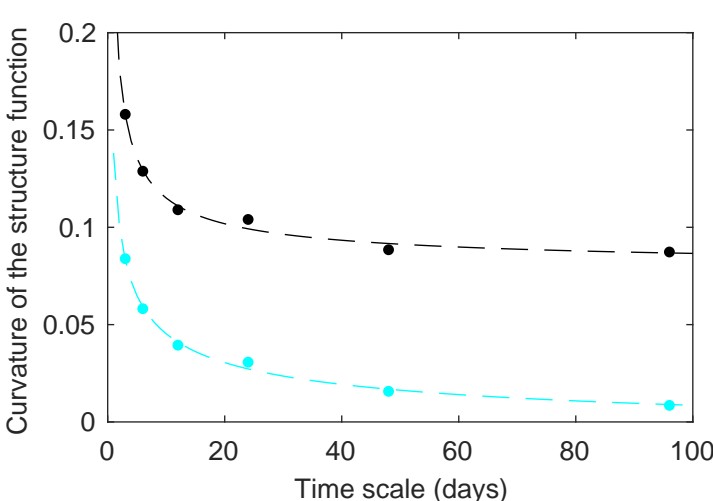

**Figure 7.** Curvature of the structure function as a function of the time scale $T$ for the model (cyan dots) and the observations (black dots). The dashed lines are power-law fits (in the least-squared sense) through the data.





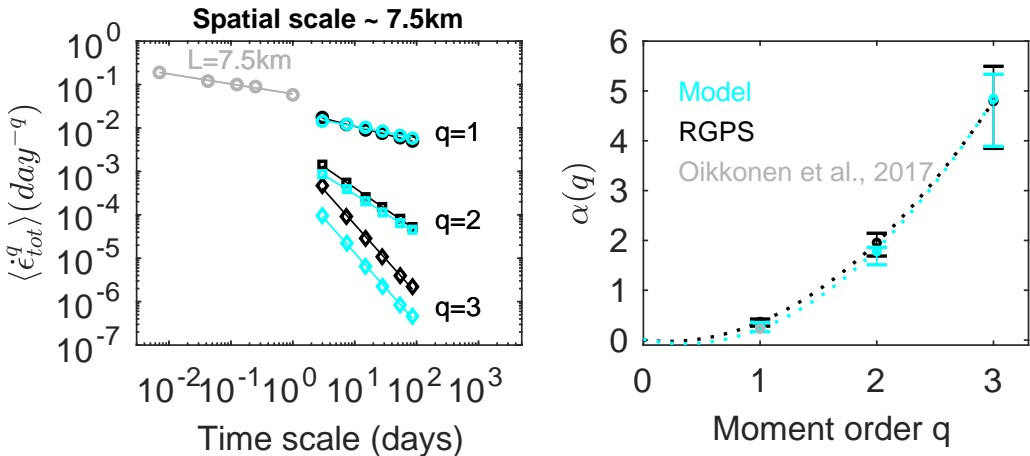

**Figure 8.** Temporal scaling analysis of the observed (black) and simulated (blue) deformation rate derived from the motion of the same triplets with initial surface area of 7.5 squared km. Left panel: Normalized moments $\langle \dot{\varepsilon}_{tot}^q \rangle$ of order $q = 1, 2$ and 3 of the distributions of the deformation rate $\dot{\varepsilon}_{tot}$ calculated at a spatial scale of 7.5 km and time scales varying from 3 to 100 days for the observations and 3 hours to 100 days for the model. The solid lines indicate the associated power-law scaling $\langle \dot{\varepsilon}_{tot}^q \rangle \sim t^{-\alpha(q)}$. The dashed lines are extrapolation for the smallest scales. Right panel: Corresponding structure functions $\alpha(q)$ for both model and observation where $\alpha$ indicates the exponent of the power laws fits, and $q$ is the moment order. The error bars are estimated from the minimal and maximal local scaling exponent as in Bouillon and Rampal (2015a) and thus correspond to upper-bound estimates.





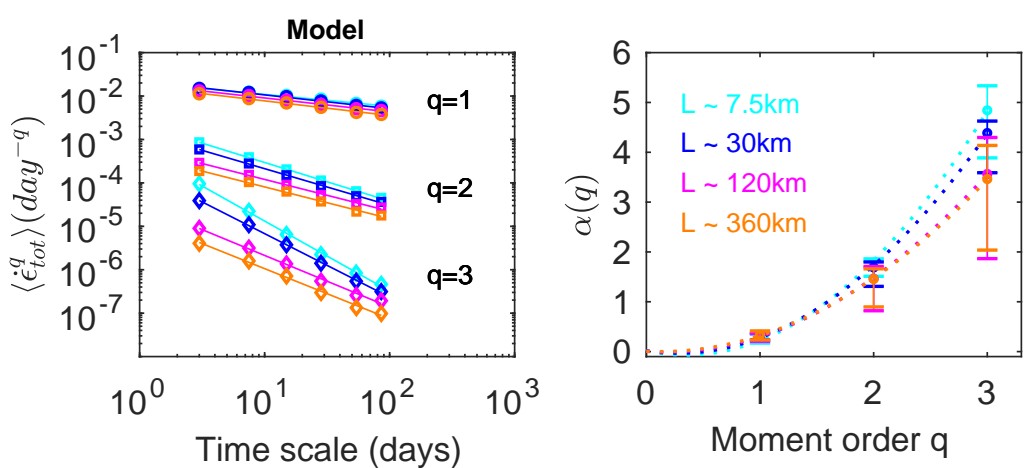

**Figure 9.** Same as Figure 8 but for the model at various spatial scales.




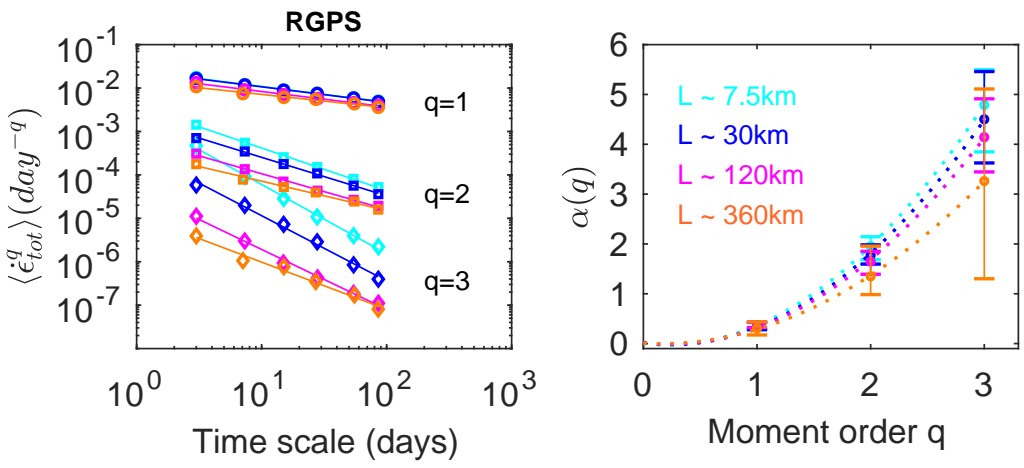

**Figure 10.** Same as Figure 8 but for the observations at various spatial scales.



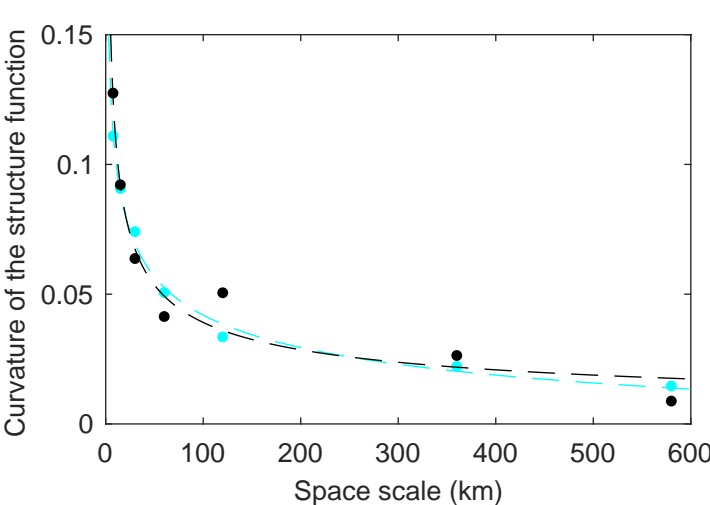

**Figure 11.** Curvature of the structure function as a function of the space scale $L$ for the model(cyan dots) and the RGPS(black dots). The dashed lines are power-law fits (in the least-squared sense) to the data.





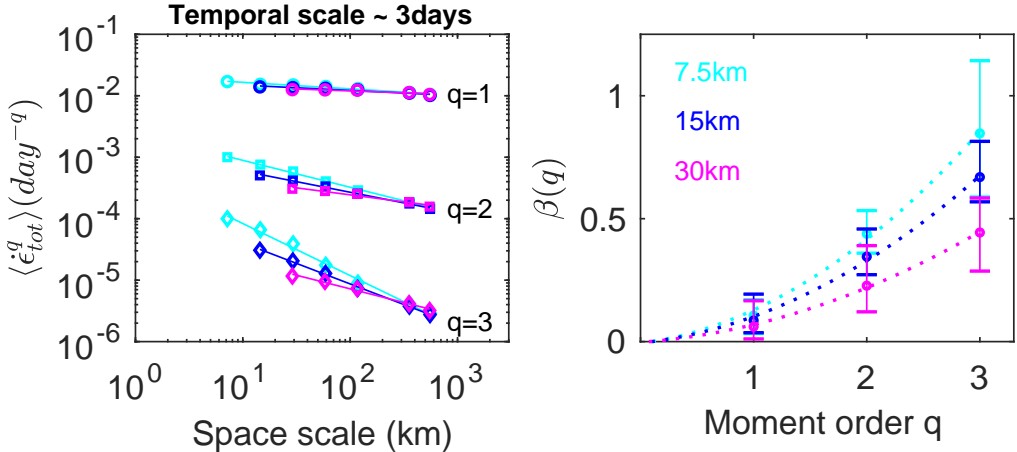

**Figure 12.** Spatial scaling analysis of the simulated deformation derived from the motion of triplets over a time scale of $T = 3$ days in 3 different model runs, at 7.5, 15 and 30 kilometers resolution respectively. Left panel: Normalized moments $\langle \dot{\varepsilon}^q_{tot} \rangle$ of order $q = 1, 2$ and 3 of the distributions of the deformation rate $\dot{\varepsilon}_{tot}$ calculated at a temporal scale of 3 days and for spatial scales varying from 7.5 to 580 kilometers. The solid lines indicate the associated power-law scaling $\langle \dot{\varepsilon}^q_{tot} \rangle \sim L^{-\beta(q)}$ as in figure 3. Right panel: Corresponding structure functions $\beta(q)$ where $\beta$ indicates the exponent of the power-law fits, and $q$ is the moment order. The error bars are estimated from the minimal and maximal local scaling exponent as in Bouillon and Rampal (2015a) and thus correspond to upper-bound estimates.



**Table 1.** List of variables used in neXtSIM.

| Symbol | Name | Meaning | Unit |
|---|---|---|---|
| $H$ | sea ice thickness | volume of ice per unit area | m |
| $h_s$ | snow thickness | volume of snow per unit area | m |
| $A$ | sea ice concentration | surface of ice per unit area | - |
| $d$ | sea ice damage | 0=undamaged, 1=completely damaged ice | - |
| $\boldsymbol{u}$ | sea ice velocity | horizontal sea ice velocity | m s$^{-1}$ |
| $\boldsymbol{\sigma}$ | sea ice internal stress | planar internal stress | N m$^{-2}$ |



**Table 2.** Parameters used in the model with their values for the simulation at 7.5 km resolution used for this study.

| Symbol | Meaning | Value | Unit |
|---|---|---|---|
| $\rho_a$ | air density | 1.3 | kg m$^{-3}$ |
| $c_a$ | air drag coefficient | $4.9 \times 10^{-3}$ | - |
| $\theta_a$ | air turning angle | 0 | degree |
| $\rho_w$ | water density | 1025 | kg m$^{-3}$ |
| $c_w$ | water drag coefficient | $5.5 \times 10^{-3}$ | - |
| $\theta_w$ | water turning angle | 25 | degrees |
| $\rho_i$ | ice density | 917 | kg m$^{-3}$ |
| $\nu$ | Poisson coefficient | 0.3 | - |
| $\mu$ | internal friction coefficient | 0.7 | - |
| $E_0$ | undamaged elastic modulus | 50.0 | MPa |
| $\Delta x$ | mean distance between mesh nodes | 10 | km |
| $\Delta t$ | time step | 200 | s |
| $c$ | cohesion | 25 | kPa |
| $\sigma_{N\,\mathrm{min}}$ | tensile strength | $-21$ | kPa |
| $\sigma_{N\,\mathrm{max}}$ | compressive strength | 75 | kPa |
| $c^*$ | compactness parameter | $-20$ | - |
| $\alpha$ | damage exponent | 4 | - |
| $\lambda_0$ | undamaged relaxation time | $10^7$ | s |
| $T_d$ | characteristic time for damaging | 20 | s |