# Peer review of "On the multi-fractal scaling properties of sea ice deformation"

_The Cryosphere, 2018_

## Referee Comment (RC1) · Anonymous Referee #1 · 28 Feb 2019

**GENERAL COMMENTS**

The manuscript applies a scaling and multi-fractal analysis to sea-ice deformation fields simulated with neXtSIM and derived from RGPS satellite observations. It shows that the spatial and temporal scaling of the observed sea-ice deformation is well reproduced by the model. The paper also explores the multi-fractality and spatio-temporal coupling of the scaling, but whether these behaviors are significant is debatable considering the very large error bars on the values compared.

The manuscript is generally well written but the wording used is often too strong or too conclusive for what the results are showing. I think the paper would be greatly improved if the authors would include a discussion of the error to support their affirmations. I also disagree with the choice of method for analyzing the RGPS data and would also

like more information on how the scaling analysis is performed. For these reasons, I recommend that is this paper be reconsidered after major revisions and I provide more details about points to be addressed in the revisions below.

Major points to be addressed:

1. I have reservations about the choices of methods used to analyze the data in this study. By using a nearest-neighbour interpolation on the RGPS trajectories, the authors artificially set all initial temporal scales in the RGPS data to 3 days, although they vary strongly from a few minutes to up to 10 days. A filtering that keep the original RGPS tempral scales, for example, keeping only the trajectories that are updated more or less every 3 days, would be more appropriate. As it is now, it is not clear to me that the temporal scaling and spatio-temporal coupling the authors are reporting here is not an artifact of the method used.

Moreover, it would be necessary to include more details on the scaling (or "coarse-graining") procedure used in this study so that the results can be reproduced by others. As of now, it is also unclear what are the effects of using a sub-sampling of the trajectories (as I understood is done) instead of spatio-temporal averaging as usually done for the scaling analysis. The differences in the method and also the justification for choosing a different method need to be clearly stated.

2. I am also not convinced of the significance of the multi-fractality and spatio-temporal coupling behavior that the authors affirm is present in the results. The error bars on the values used to infer these behaviors are sometimes used to confirm that values overlap and therefore are equal, but are elsewhere ignored to affirm that the values are different. Also, the "error bars" as defined in this study rather represent the goodness of fit on the data, than an actual error on the values calculated. Proper error estimates are needed to claim that the multi-fractality or the coupling are significant.

3. The discussion of scaling in sea-ice models is limited to (Maxwell) Elasto-Brittle (EB/MEB) papers. There are several studies using the viscous-plastic (VP) rheology

that are worth mentioning. Especially, the claim that for the first time spatio-temporal scaling is shown for a model, is not true (see Fig. 7 in Hutter et al., 2018 - see reference below).

4. The conclusion is mostly a summary of what the neXtSIM model is capable of rather than overall conclusion that can be drawn from the presented work for other studies or model development. The overall conclusion that is drawn (that is, that multi-fractal scaling analysis should be the prerequisite validation step before analysis of any other variable that might be related to sea-ice dynamics) is clearly application-dependent and needs to be modified. I would wish that the authors come up with a conclusion that is more useful for the scientific community than just promoting the model. There is clearly enough good material in the paper to do so.

5. I would appreciate if the model physics and the configuration of the simulation (e.g. wind forcing, rheology parameters, Lagrangian grid, etc.) would be separated more carefully when drawing conclusions from the simulated results. It feels like when there is agreement between the model and observations, the authors attribute this agreement to their choice of model physics, while if the model results disagree with the observations, the authors note that it could be due to the model configuration. Maybe a change in wording would help to reduce this impression.

Below are more specific comments to help the authors address the general comments above.

==================================================================
SPECIFIC COMMENTS

Page 2: - Line 16: There is a new paper about filtering LKFs in the entire RGPS data-set that also shows are wide range of intersection angles of LKFs (Hutter et al., 2019, see References below) and is worth to citing here.

Page 3: - Line 4: Again, please add Hutter et al. (2019) (Linow and Dierking only

studied 10 RGPS snapshots, whereas in the above mentioned paper, the LKF length was studied for the entire RGPS data-set).

- Line 7: "These events also sustain deformation, maintaining the LKFs "active" for many days Coon et al. (2007)" This is not clear. If the deformations are of short duration, how are they responsible for "sustaining" deformation rates over many days? Please clarify.

- Line 15: To me, it is the distribution that can be in or out of the Gaussian basin of attraction, not the values themselves. I would remove "that are out of the Gaussian basin of attraction" or rewrite something like "...i.e., dominated by extreme values and out of the Gaussian basin of attraction"

- Line 25: Please shortly explain what a "coarse-graining" method is.

Page 4: - Line 6: I agree that beta=0 is homogeneous deformations. But if one imagines the scaling exponent to be something similar as the fractal dimension (see Weiss, 2003), then beta=1 would correspond to deformations concentrated in one line, and for beta=2, all deformations would be localized in one single point. This is also what one would expect from averaging in two spatial dimensions. Please clarify.

- Lines 21-24: Two paragraphs above you state that the distribution of the sea-ice deformation rates is out of the Gaussian basin of attraction (i.e. the decay has a slope <= 3). For those power-law distributions it is known that the higher moments (variance, skewness, etc.) do not converge, due to the presence of extreme values, to a real number but to infinity. Please clarify how those ill defined moments help better describe the distribution?

Page 5: - Line 5: "the amount of localization of large and small deformation events is the same" How do you define the localization? I thought the scaling exponents are quantifying the "degree of localization" of deformation (see page 4, line 5)? Here, even if the curvature is zero, the scaling exponents are still changing linearly with the

moment q, so that they are different and then I don't think you can say that the "amount of localization of large and small deformation events is the same".

- Line 10-11: Please add references for this. The definitions of heterogeneity and intermittency are confusing to me... Isn't saying that a field shows high localization in space/time (ie. beta or alpha > 0) the same as saying that the field is heterogeneous/intermittent? Why does a field need to have a (quadratic) change in the localization exponent with the different moments q to define it as heterogeneous/intermittent? What if the structure function is cubic? Please clarify.

- Line 20 - "(Olason et al., 2019)" I couldn't find this paper in the Cryosphere Discussion.

- Line 24,25: Downscaling of the modeled sea-ice deformation might be performed if one characterizes the scaling exponents, however, Spreen et al. (2017) have shown that the dependence of the scaling exponent on the sea-ice concentration and thickness has non-trivial effects on the "scaled" deformation rates, so I would suppose that only if you knew the distribution of A and h at a subgrid-scale (which we don't) could you actually perform such a downscaling.

Page 6: - Lines 3-6: It would be worth to include a short summary of studies that used scaling analysis to evaluate sea-ice deformation in simulations and their findings (so far only EB studies are mentioned). For example, Hutter et al. (2018) have shown that the VP rheology can reproduce the observed spatial scaling and also multifractal characteristics in the spatial domain (i.e. quadratic spatial structure function at very high resolution. Please see also Spreen, et al. (2017) and Bouchat and Tremblay (2017).

- Lines 16-19: Please elaborate on why this is needed. What does it mean "to localize the deformation at the nominal scale"?

Page 9: - Line 2: Please provide the spatial and temporal resolution of the forcing used.

Page 10: - Line 12: "We use the coarse-graining approach..." Since you already mention both approaches, please justify why you haven chosen the first one.

- Line 13: Why do you choose triplets? It is known that the boundary definition error (Lindsay & Stern, 2003) is larger for lower number of vertices. You have chosen the minimal number of vertices and thus the highest uncertainty. Why? Do you use the smoothing filter as suggested by Bouillon and Rampal (2015) in your analysis to compensate this effect? If not, please mention why not and how you deal with the uncertainties introduced by choosing triangles instead of rectangles as done originally in RGPS.

- Line 15: The value of 7.5 km is an average for the triangulation over the 2006/2007 season? Otherwise, it is not clear to me how you get that number analytically. Please specify.

- Line 25: Do you use the same triangulation for both the RGPS and model trajectory sets? Also, how do you handle the different streams of the original RGPS Lagrangian ice motion dataset?

Page 11: - Line 5-8: Not clear. What is the "subsampled cloud" of positions? How do you select the triangles to add up to a certain spatial scale? Please add details of the subsampling procedure!

- Line 9: "The number of triplets available for the statistical analyses decreases as the space scale increases." Why? Because you use a filter to discard larger scales if they are not filled up to a certain percentage? Please clarify.

- Line 9-11: "Coarse-graining in time..." Are you re-sampling the trajectories at larger time intervals and then computing new estimates of the strain rates at these larger time scales, instead of averaging multiple 3-day strain rates together? To be consistent with the spatial scaling analysis as it is usually done (e.g. as in Marsan et al. 2004), the strain rates should be averaged and not re-sampled.

- Line 10: "The number of available triplets also decreases as the time scale increases."

[Figure]

Again, please indicate why this is the case.

- Line 18: "...around the boundary of each polygon associated to a given space scale L" Again, it seems like you are saying that you are re-calculating the strain rates at different scales instead of averaging multiple triangles of the original data set together to add up to a certain time/spatial scale. If you are really recalculating the strain rates instead of averaging them, then please show what are the effects of doing this vs averaging on the scaling analysis, as I don't recall other studies that have used this method. Or maybe simply re-calculate your strain rates by averaging the triangles instead.

Page 12: - Line 1: The actual area A of the polygons will differ slightly from the nominal scale $L^2$. Do you filter the polygons for their area to match the given nominal length scale? Also, are you using the definition of A with the summation around the vertices of the polygons, e.g. as in Linsday and Stern (2003)? Please specify.

I am also confused about which polygons are which... If you are always grouping triangles that have a mean length scale of $L_i = 7.5$ km to make new polygons at different length scales, wouldn't the average length scale defined as "the mean of the square root of the polygon surface areas", as written on page 11, i.e. what I understand as $L = (L_1 + L_2 + L_3 + ... + L_n)/ n$ (with $L_i$, i=1,...,n being the length scale of the individual triangles), also equal about 7.5 km regardless of the number of triangles you average together? I feel like it would make more sense to define the spatial scale L of the new polygons (i.e. the aggregation of triangles) as the square root of the sum of all the triangle areas in the polygon, i.e. $L = \sqrt{A_1 + A_2 + A_3 + ...}$ where $A_i$ is the area of the i-th triangle that is averaged together. Or maybe I just don't understand your scaling procedure. Please clarify.

Page 13: - Line 1: I strongly disagree with this procedure. A nearest neighbour interpolation will artificially set all initial temporal scales in RGPS data to 3 days, although they vary strongly from a few minutes to up to 10 days. Why do you not use the original

temporal scale of the observations for the scaling analysis? How much of the method is therefore responsible for the temporal or spatio-temporal scaling you are showing afterwards?

- Lines 10-15: Because trajectories are eventually removed from your analysis by filtering? Or why else is this the case?

- Line 17: "the 3-day shear [...] for the same period of 7 days" How do you get the strain rates on a 7-day period if they are the 3-day strain rates?

- Line 19: Technically, what you are showing is not the cumulative probability distribution (CDF), but the complementary cumulative distribution function (CCDF), i.e. the probability of having a value greater than a given strain rate. Please correct. Also, why choose to show the CCDF instead of the PDF as in previous studies? It would be interesting to show here the PDFs of shear and divergence since it is the first time it would be shown for the MEB rheology in this configuration.

- Line 20-21: Please discuss the fact that the probability distribution for your model is always greater than that of RGPS. What does this imply?

- Line 21: If we assume a power law probability distribution function (PDF) that goes like $P(x) \rightarrow x^{-alpha}$, then the CDF (or CCDF) would decay like $C(x) \rightarrow x^{-alpha+1}$. Hence, if you find a slope of -3 for the CCDF of both your model and RGPS, it means that the PDFs for both data sets decay with a slope of -4, which according to Sornette (2006), implies that the PDFs slowly converge to Gaussian distributions (or that they are in the "Gaussian basin of attraction") and therefore your argument following in the text does not hold... Please address this.

Page 14: - Line 3: Stern et al. (2018) suggest to use Maximum Likelihood Estimators to determine power-law exponents and test those with a goodness-of-the-fit test (Clauset et al., 2009).

- Lines 18-19: Defined as in Bouillon and Rampal (2015), these bars are rather representing the "goodness of the linear fit" rather than an actual error on the values you are comparing. It would be much more useful (in terms of comparing the model to observations) to compute the error on your observed and simulated deformation rates given the known error on the trajectory positions (see for example Lindsay and Stern, 2003) and then the ensuing error on your scaling analysis. Only then can you conclude that the structure functions are "equal within their margin of error".

- Line 21: "... the scaling is clearly multi-fractal, as no linear function can be contained within the error bars." I can pass a line through the origin and through all the "error bars" for the model values. Please remove.

- Line 22: "applying a quadratic fit" Please provide the quadratic fit parameters for both RGPS and the model, either here or on the figure.

Page 15: - Line 12-13: Mean curvatures of 0.07 and 0.08 seem quite low to consider this a "clear" signature of multi-fractality... Again, it would be necessary to have the error on these values to know if it is significant or not.

- Line 16: "beta decreases with increasing T" This is not very clear from figures 5 and 6... In fact, from the right panel in figure 5, it looks more like beta is increasing with increasing T for q=1. Please add a log-log plot of beta vs T for the different values of q for both RGPS and the model, similar to what is done in figures 5 and 7 in Hutter et al. (2018).

- Line 20: "This property is for the first time shown from a sea ice model simulation." This is not true. See Figure 7 in Hutter et al. (2018). Moreover, is this coupling really significant, as the "error bars" overlap for all temporal scales (for each moment respectively)? If you say that the structure functions for both RGPS and the model are equal in Figure 3, then I would also say they are equal here in Figure 5 for all time scales, and we therefore cannot conclude to a significant coupling.

Page 16: - Lines 3-4: A few more words might be helpful here to understand this offset

in the model curvature: Is MEB leading to damage in the ice cover everywhere and, therefore, evenly distributed events with no preferred regions of deformation?

- Line 20: "This means that the proportion of extreme deformation events compared to lower ones is too small or that their values are too low in the simulation." The CCDF for the shear deformations in Figure 1 actually shows that the probability of having larger deformations is higher in the model than for RGPS, no? If you show the PDFs of shear and divergence, it would probably help to clarify this.

Page 17: - Line 12-15: It is not clear from Figures 9 and 10 that alpha is decreasing for increasing L. Please add a plot of alpha vs L for q=1,2,3. Again, the question of whether this coupling is significant if all alpha(q=1,2) lie within the errorbars of alpha(q=1,2) arises.

- Line 23: "reproduces correctly the distribution of sea ice deformation rates" Please show PDFs of shear and divergence to affirm this.

Page 18: - Line 9: "a threshold mechanism" Is that the damage parametrization?

- Line 14: You show that your model reproduces some of the observed scaling characteristics, but you have not shown that it does because your model includes the "ingredients" mentioned above. The configuration of the model (i.e. forcing, strength parameters, etc.) as well as the Lagrangian mesh instead of an Eulerian grid also have the potential for generating/influencing these behaviors, and it is not clear yet to which model parametrization or configuration ingredients these behaviors are due.

- Line 16: "the spatial scaling [...] holds down to the nominal resolution of the mesh" and just after "It means that neXtSIM does not need to be run at higher spatial resolution in order to resolve the presence of linear kinematic features..." I am not sure that the first sentence justifies the second one... For example, Spreen et al. (2017) and Bouchat and Tremblay (2017), both show that VP models at 9 km and 10 km can also have a spatial scaling that "holds down" to 10 km (ie. the nominal resolution), but you would

still need to run the model at higher resolution if you want to resolve finer structures in the sea-ice fields because the models are represented on Eulerian grids. In the case of your model, you might better resolve LKFs because you are using a Lagrangian mesh, which represents discontinuities more accurately, not necessarily because of the scaling of deformations.

- Line 20: Add reference to Hutter et al. (2018)? You seem to be indirectly referring to this study.

- Lines 21-28: This should be moved to the results section.

Page 20: - Line 12-13: I disagree. See comment for Page 15, Line 20.

- Lines 21-25: This is a too strong statement that depends a lot on what the model is used for. There are applications where heterogeneity and intermittency of deformation are important (i.e. regional and short range forecasting of ice conditions) but there are also larger scale applications where other parameters are more relevant. Either remove this statement or give specific application areas where this is needed.

Page 22: - Line 10: In Dansereau et al. (2016), d=1 for undamaged and d=0 for completely damaged ice. Please indicate that you use the reverse definition.

Page 23: - Lines 11-13: g(H) is not defined.

- Equation (A13) and (A14): The prime variables have not been defined. in (A14), shouldn't it be sigma_1_prime and sigma_2_prime instead?

Page 24: - As it seems that the implementation of this 3-thickness categories differs from Stern and Rothrock (1995), I would like to have a bit more details on how it is done/defined and how it is different from what was suggested Stern and Rothrock (1995). For example, please explain the addition of the divergence term in the evolution equations and a term for ridging for the thin ice category as well. Please also clarify if A and H are the total ice concentration and volume per unit area? i.e. A = A_thin + A_thick and H = H_thin + H_thick?

- Lines 16-18: "Thin ice thickness is considered to be uniformly distributed between hmin and hmax", do you mean linearly distributed? Why does that put a maximum bound on the total ice volume per area? Maybe here it should be "H_t_min = A_t * h_min" and "H_t_max = A_t * (h_min+h_max)/2" instead?

Page 26: - Equation (A23): Please explain why you introduce this delta_A variable and what is the purpose of the "aspect ratio parameter" zeta, and what a value of 10 implies.

- Line 8: Shouldn't more ridging also affect the value of Hˆ{n+1}?

Figure 1: - I would switch for PDFs and also add a panel with divergence distributions.

Figure 2: - Please also show the divergence fields.

Figure 3: - The left panel y-axis shows the scaling for eps_tot, however, in the caption it is written that you are showing the scaling and the structure for the shear deformation rate... Please show the scaling and the structure function for eps_tot instead.

Figures 5 & 6: - You could group these two figures for ease of comparison between the model and RGPS observations.

Figure 8: - Caption, Line 3: Normalized moments have not been defined in the text.

Figures 9 & 10: - You could group these two figures for ease of comparison between the model and RGPS observations.

====================================================================
TECHNICAL CORRECTIONS

Page 2: - Line 7: - Delete "for the first time" - Line 16-17: "e.g.," should come before enumerating the references.

Page 3: - Line 2: Replace "kinematic linear features" with "Linear Kinematic Features" - Line 6: Delete "levels of" - Line 7: "Coon et al., 2007" should be in parenthesis -

Line 9: Add a coma after (Kwok, 2001), ie: "(Kwok, 2001), and permanent..." - Line 13: Delete "such as" and replace with "... of the deformation rate invariants (i.e. shear and divergence) and of the total deformation rates, which..." - Line 26: I think there is a part missing in this sentence. Maybe add "applied to observed deformation fields derived from satellite imagery" before "(e.g. Lindsay..." or something like that? - Line 27: Replace "or pair of buoys dispersion analysis" with "or by dispersion analysis of pair of buoys"

Page 4: - Line 4: Add "The scaling exponents..." - Line 9-10: Rewrite "...to a homogeneous deformation, and alpha=1 to a single, temporally isolated deformation event." - Line 16: "approximated" –> "modeled as" and "relevant for Arctic system simulations"? - Line 18: Delete "out of the Gaussian basin of attraction" (see specific comment for Page 3, Line 14). - Line 25: Replace "beta" with "the scaling exponents beta and alpha" - Line 27: Delete "indeed"

Page 5: - Line 4: "... linear structure function, i.e., no curvature, ..." replace with "... linear structure function, i.e. no curvature or equivalently a=0 or b=0, ..." - Line 6: Replace "For both coefficients..." with "In the case where both coefficients..." - Line 7: Add "therefore" between "distribution" and "increase", i.e "...of the distribution therefore increase..." - Line 12-13: Replace "have shown" with "show"

Page 6: - Line 7: "... in the deformation and related characteristics of sea ice" Not clear. Please reformulate. - Lines 21-23: This sentence is not clear. Please reformulate.

Page 7: - Line 1: Replace "the first part of the paper" with "Section 1 and section 2 of the paper" and delete "(section 2)" in line 2. - Line 2: Replace "The second part" with "Section 3" and remove "(Section 3)" in line 4. - Line 10: Replace "(Amitrano et al., 1999)" with "Amitrano et al. (1999)" - Line 17: Rewrite "In particular, it was shown that the simulated deformation rates..." - Line 18: Add "... in space only".

Page 8: - Line 8: Replace "entering" with "of" - Line 9: Replace "Appendix" with "appendices" - Line 15: Replace "length of the vertices" with "distance between the vertices"

- Line 24: Remove "the applied", and all of "the" in front of the quantities enumerated

Page 10: - Line 1: Replace "displacement" with "ice motion"

Page 12: - Line 18: Add "... 30 degrees or less". - Line 19: Replace "as the model is" with "contrary to the model" - Line 23: "affect" should be "affects" - Line 23: "sub-sampling" Do you mean interpolation?

Page 13: - Line 6: "we therefore chose to..." You do not chose, you can't go below 3 days given that this is the smallest time scale you have for your dataset. - Line 7: Remove "on the whole" - Line 17: Maybe relabel Figure 2 to Figure 1 since you are referring to it first? - Line 17: Replace "3-days" with "3-day"

Page 14: - Line 2: Add "... spatial scaling analysis for a T= 3 days temporal scale..." - Line 12: Add "our choice of mechanical parameters values (e.g. Bouchat and Tremblay, 2017)" - Line 22: Replace "a quadratic fit to the data (in the least squared sense)" with "a least-square quadratic fit to the data" - Line 25: Add "... simulated deformation fields is consistent..." - Line 27: Add "... the value of the spatial scaling exponent beta..." - Line 27: "for the mean obtained for the successive and contiguous snapshots throughout the winter" This is not clear. The mean = mean deformations , i.e. q=1? Please rewrite.

Page 15: - Line 1: Replace "the scaling exponent varies" with "the spatial scaling exponent varies" - Line 4: Replace "for the mean" with "for the mean deformation rates (i.e. q=1)" - Line 5: Add "... which is also the value..." - Line 10: Add "...that the observed and simulated curvature values..." - Line 21: "The origin of this coupling has been previously proposed to be linked to the complex correlation patterns related to chain triggering of ice-quakes." Please add reference for this. - Line 24: Add "... the multi-fractal character of the spatial scaling (i.e. the curvature of beta(q)) for both RGPS and the model when..."

Page 16: - Line 9: Remove "robust" and "very similar" since you then discuss how it

differs for the q=3. - Line 16: Replace "in this recent study" with "by Oikkonen et al. (2017)" - Lines 17-18: Remove "(gray, dark and cyan top curves in the left panel of Fig. 8)"

Page 17: - Line 4: "virtually perfect" please change to a more sober wording. For example, the values for q=2 are not "perfectly" matching. - Line 4: Rewrite "The curvature of the quadratic functions alpha(q) are 0.11 for..." - Line 7: "This seems to argue that..." Weird wording. Please rephrase.

Page 19: - Line 18: Replace "." after "distribution" by a coma, and change "A proper..." for "a proper..." - Line 24: Replace "concurrent" with "parallel"?

Page 20: - Line 2: Add "from RGPS observations..." - Line 13: Remove "for the first time" and "by a model"

Page 21: - Line 16: Replace "thick ice thickness" with "thick-ice thickness"

Page 22: - Line 15: Why not write -c* with c* =20 as done in Hibler (1979)? And you could put (A6) back in (A5) to save space.

Figure 1: - Please add legend in the figure for ease of comparison - Add the fit exponents on the figure or in the caption, for both model and RGPS. - Caption, Line 2: Add "...and the RGPS observations"

Figure 2: - There seems to be a plotting issue since some of the triangles are touching the land boundaries (e.g. on the Alaskan coast), but you mention in the manuscript that you filter out trajectories that are 100 km or closer to land. Please correct. - Please add "RGPS" and "Model" on top of the panels. - Caption: Add that the green lines are the model's open boundaries.

Figure 3: - Caption, Line 2: Replace "than in the RGPS dataset" with "and RGPS dataset" - Caption, Line 7, "local scaling exponents" Not clear. Use a similar wording as in Bouillon and Rampal (2015). - Caption: Use the same wording for the caption as for Figure 8 (with the suggested corrections).

Figure 4: - Please add legend in the figure for ease of comparison - Caption, Line 1: Replace "power scaling exponents" with "spatial scaling exponents for the average total deformation (i.e. q=1)" - Caption: Add something like "calculated for individual snapshots, i.e. at a temporal scale of T = 3 days"

Figure 5: - Please use same y-axis for left panels in Figures 3,5,6,8,9,10,12 for ease of comparison. - Please use same y-axis for right panels in Figures 3,5,6,12

Figure 6: - Caption: Add "..for the RGPS observations..."

Figure 8: - Caption, Line 3: Replace "distributions of the deformation rate" with "distributions of the total deformation rate" - Caption, Line 4: Switch "for the observations" with "for the model" later in the sentence, and indicate that the values for T=3hrs to 1 day are taken from Oikkonen et al. (2017) in the caption as well. - Caption, Line 6: Remove "The dashed lines are extrapolation for the smallest scales" There are no dashed lines. - Caption, Line 7: Replace "observation" with "RGPS observations"

Figure 10: - Caption: Add "..for the RGPS observations..."

==================================================================
REFERENCES

Bouchat, A., and B. Tremblay (2017), Using sea-ice deformation fields to constrain the mechanical strength parameters of geophysical sea ice, J. Geophys. Res. Oceans, 122, doi:10.1002/2017JC013020.

Clauset, A, Shalizi, CR and Newman, MEJ. 2009. Powerlaw distributions in empirical data. SIAM Rev 51(4): 661–703, https://doi.org/10.1137/070710111

Hutter, N., Zampieri, L., and Losch, M.: Leads and ridges in Arctic sea ice from RGPS data and a new tracking algorithm, The Cryosphere, 13, 627-645, https://doi.org/10.5194/tc-13-627-2019, 2019

Hutter, N., Losch, M., & Menemenlis, D. (2018). Scaling properties of arctic sea

ice deformation in a high‐resolution viscous‐plastic sea ice model and in satellite observations. Journal of Geophysical Research: Oceans, 123, 672–687. https://doi.org/10.1002/2017JC013119

Lindsay, R.W. and H.L. Stern, 2003: The RADARSAT Geophysical Processor System: Quality of Sea Ice Trajectory and Deformation Estimates. J. Atmos. Oceanic Technol., 20, 1333–1347, https://doi.org/10.1175/1520-0426(2003)020<1333:TRGPSQ>2.0.CO;2

Sornette, D.: Power Law Distributions, pp. 93–121, Springer Berlin Heidelberg, Berlin, Heidelberg, doi:10.1007/3-540-33182-4_4, https://doi.org/10.1007/3-540-33182-4_4, 2006.

Spreen, G., Kwok, R., Menemenlis, D., and Nguyen, A. T.: Sea-ice deformation in a coupled ocean–sea-ice model and in satellite remote sensing data, The Cryosphere, 11, 1553-1573, https://doi.org/10.5194/tc-11-1553-2017, 2017.

Weiss, J. Surveys in Geophysics (2003) 24: 185. https://doi.org/10.1023/A:1023293117309

---

## Referee Comment (RC2) · Anonymous Referee #2 · 6 Mar 2019

General comments: This paper aims at validating the basic behavior of deformation rate in the neXtSIM sea ice model which is based on the Maxwell-Elasto-Brittle rheology, focusing on the scaling properties in space and time. The model domain was the whole Arctic Ocean and the coarse graining method was used for scaling analysis with the drifters' data in the model. For validation data, the Lagrangian displacement data produced from RADARSAT Geophysical Processor System (RGPS) were used. Through scaling analysis, it was shown that the multi-fractal properties can be reproduced for the first time in the numerical sea ice model. Besides, the statistical properties of the first, second, and third moments of deformation rates at temporal scales of 3 days to 96 days and spatial scales of 7.5 km to 700 km were shown to be mostly consistent with the observations. In conclusion, since the fundamental properties were

validated, they suggest that the neXtSIM model could be used as a proper tool to further study the physical meaning of the processes related to deformation. Considering that it is still a big challenge to reproduce the rapid thinning trend of ice thickness distribution in the Arctic Ocean in the numerical sea ice model and the need to improve the deformation processes in the model has been recognized for a long time, the topic of this paper is timely, and the results of this paper will provide insightful implications. Overall, I feel that this paper is an elaborate and nice work, and this approach is indispensable to improve our understanding of the dynamic behavior of sea ice. Therefore, I believe this work will contribute to the development of sea ice dynamics, especially for the parameterization of the model, related to the deformation. My comments, which might come from the lack of my knowledge about mathematics, are limited to minor points as follows:

1) Regarding the description of exponents, $\alpha$ and $\beta$ (Eqs. 4 and 5), could you please explain more about why these exponents can be expressed as a quadratic equation of the moment parameter (q)? To be honest, I could not follow the subsequent paragraph (P5L4-11) completely. To my understanding, multi-fractal means the geometric properties that contain various dimensions of fractals. If this is correct, why can the curvature of the exponents as a function of q be an indicator of multi-fractal which discriminates from mono-fractal? In my mind, if I could accept this concept, the manuscript would have become much more understandable to me. 2) Regarding the methodology of analysis, it is stated that you used the coarse-graining approach (P10L12). Is this the method described after P11L12? If so, it might make it readable when you insert "(shown later)" at the end of the sentence (P10L12). Besides, regarding the statement, "Only the trajectories that are common to both the simulation and RGPS dataset are considered in the calculation of the deformation and their statistics" (P10L22-24), I am a bit concerned whether this approach might affect the results by setting a bias in the calculation. I mean the data consistent with observations might have preferentially selected. If you can add some description about how much fraction of data were discarded by this method and show that this selection did not affect the result

significantly, it would be appreciated. 3) Regarding the interpretation of the scaling analysis (Fig.5&6), it is stated that "We find that the estimated spatial scaling exponent, $\beta$, decreases with increasing T (Figure 5 and 6, left panels)" (P15L15-16). To my understanding, $\beta$ corresponds to the slopes of the graphs. As far as looking at the left panels, however, the slopes appear not to be significantly different for all the values of T (3 days to 96 days) at least for q = 1. When looking at right panels, there certainly be a decreasing trend with the increase of T for q = 2 and 3. Thus, unless there is a physical meaning in the decreasing trend of $\beta$ with the increase of T, it might be one idea to focus on the decrease of the multi-fractality of the spatial scaling with the increase of T. The similar discussion may apply for the last paragraph in section 4.2 (P17L11-21). Besides, the additional description about the physical implications of the decrease of the multi-fractality would be appreciated if it is possible.

Specific comments: *(P2L19-20) "Rothrock and Thorndike, 1984; Matsushita, 1985" & "Rothrock and Thorndike, 1980" are missing in the reference lists. *(P3L7-8) "Coon et al. (2007)" should be "(Coon et al., 2007)" *(P12L18) Is there any meaning in the selection of 30 degrees? *(Figure 1&2) Considering the order of appearance in the manuscript, it would be preferable to exchange Figure 1 and 2. *(P15L4) I think "0.2" should be "-0.2". *(Figure 8) It is stated that "The dashed lines are extrapolation for the smallest scales" in the caption. However, I could not see the dashed lines. Besides, "L=7.5km", which appears in the upper left corner of the figure, is misleading. Please take it if not necessary.

That is all. Faithfully yours.

---

## Author Comment (AC1) · 18 Jun 2019

The comment was uploaded in the form of a supplement:
https://www.the-cryosphere-discuss.net/tc-2018-290/tc-2018-290-AC1-supplement.zip

---

## Author Comment (AC2) · 18 Jun 2019

The comment was uploaded in the form of a supplement:
https://www.the-cryosphere-discuss.net/tc-2018-290/tc-2018-290-AC2-supplement.zip

---

## Author Response (AR1)

Manuscript prepared for The Cryosphere Discuss. with version 2014/09/16 7.15 Copernicus papers of the LATEX class copernicus.cls. Date: 18 June 2019

**On the multi-fractal scaling properties of sea ice deformation**

Pierre Rampal1, Véronique Dansereau1, Einar Olason1, Sylvain Bouillon3, Timothy Williams1, Anton Korosov1, and Abdoulaye Samaké2

[revised manuscript text omitted]

 (Mohr-Coulomb criterion), (A10)

$$-\frac{\sigma_1 + \sigma_2}{2} \le \sigma_{T \max} \underline{g(H)} \quad \text{(tensile stress criterion)}, \tag{A11}$$
$$\frac{\sigma_1 + \sigma_2}{2} \le \sigma_{N \max} \underline{g(H)} \quad \text{(compressive stress criterion)}, \tag{A12}$$

where  $q = \left[ \left(\mu^2 + 1\right)^{1/2} + \mu \right]^2$ ,  $\sigma_c = \frac{2c}{\left[ (\mu^2 + 1)^{1/2} - \mu \right]}$ ,  $\mu$  is the internal friction coefficient, c is the cohesion,  $\sigma_{T \max}$  is the maximal tensile strength and  $\sigma_{N \max}$  the maximum compressive strength (see table 2). The cohesion, *c*, is scaled as a function of the model spatial resolution, as described in Bouillon and Rampal (2015a).

When one of the damage criteria is met, d is modified according to by multiplying (1-d) with  $\Psi_{1}$  or

$$\quad 1 - d' = d \leftarrow 1 - \Psi(1 - d), \tag{A13}$$

where

10

$$\Psi = \begin{cases} \frac{\sigma_c}{\sigma_1 - q\sigma_2} & \text{if } \sigma_1 - q\sigma_2 > \sigma_c \\ \frac{2\sigma_T \max}{-\sigma_1 + \sigma_2} & \text{if } -\frac{\sigma_1 + \sigma_2}{2} > \sigma_T \max \\ \frac{2\sigma_N \max}{\sigma_1 + \sigma_2} & \text{if } \frac{\sigma_1 + \sigma_2}{2} > \sigma_N \max. \end{cases}$$
(A14)

Healing is included here to represent the counteracting effect of refreezing of water within leads on the level of damage of the ice cover. It is implemented via a constant term in the damage evolution equation:

$$\frac{Dd}{Dt} = \frac{(1-d)(1-\Psi)}{T_d} - \frac{1}{T_h},$$
(A15)

where  $T_h$  is the characteristic time for healing and  $T_d$ , the characteristic time for damaging (Dansereau et al., 2016).

**A2 Ice thickness redistribution and thermodynamics**

- neXtSIM includes the a multi-category model inspired from Stern and Rothrock (1995), i.e. considering 3 ice categoriessuggested by Stern and Rothrock (1995) categories: thick ice, thin ice and open water. In our implementation the thin ice is only newly formed ice, so ice will only be transferred from the thin-ice category to thick ice, but not in the reverse direction. In addition, we don't apply additional open water source terms, and nor do we use the formulation of Gray and Morland (1994) to keep the ice concentration less than 1.
  - (We simply redistribute ice and snow volume if this occurs.) Thin ice is described by its

concentration,  $A_t$ , and volume per unit area,  $H_t$ , and snow volume per unit area,  $h_{s,t}$ . Thick ice is described by the concentration, A, and volume per volume per unit area,  $h_{s,t}$ . Thick snow volume per unit area,  $h_s$ . We assume that the thin ice has no mechanical strength and simply follows the motion of the surrounding thick ice.

5 Note the total ice concentration and volume per unit area are  $A + A_t$  and  $H + H_t$ , and the total snow volume per unit area is  $h_s + h_{s,t}$ .

Thin ice thickness is considered to be uniformly distributed with thickness  $h_t = H_t/A_t$ required to be between  $h_{min} = 5$  cm and  $h_{max} = 50$  cm so that the volume per unit area is bounded between  $H_{min} = Ah_{min}$  and  $H_{max} = A \frac{h_{min} + h_{max}}{2}$ . cm and  $h_{max} = 27.5$  cm. The evolution equations for A, H,  $h_{s,r}A_t$  and  $h_{s,t}$  have the following form:

 $\frac{D\phi}{Dt} = -\phi \nabla \cdot \boldsymbol{u} + \Psi_{\phi} + S_{\phi},\tag{A16}$

where  $\frac{D\phi}{Dt}$  is the material derivative that is defined for any scalar and vector as as

$$\frac{D\phi}{Dt} = \frac{\partial\phi}{\partial t} + (\boldsymbol{u} \cdot \nabla)\phi.$$
(A17)

Here  $\nabla \cdot \boldsymbol{u}$  is the divergence of the horizontal velocity,  $\Psi_{\phi}$  a sink/source term due to ridging, and  $S_{\phi}$  a thermodynamical sink/source term. Volume conservation is imposed by setting  $\Psi_{H} = -\Psi_{Ht}$  and  $\Psi_{H} = -\Psi_{Ht}$  and  $\Psi_{h_{s}} = -\Psi_{h_{s,t}}$  and an additional constraint is that  $A_{h} + A \leq 1A_{t} + A \leq 1$ .

The evolution of A, H,  $A_t$  and  $H_t$  is computed following three main steps (variables updated in each step are denoted with a prime):

20 1. Advection: The scheme solves the equation:

10

$$\frac{D\phi}{Dt} = -\phi \nabla \cdot \boldsymbol{u},\tag{A18}$$

for each conserved scalar quantity (A, H,  $A_t$ ,  $H_t$ , etc.). For this paper, we use the purely Lagrangian scheme presented in Rampal et al. (2016). After this step the concentration could be larger than 1.

- 2. Mechanical redistribution: The scheme imposes the limit  $A_t + A \le 1$  on the total ice area by following those steps:
  - (a) Compute the new open water concentration as:

$$A_0 = \max(0, 1 - A - A_t),; \tag{A19}$$

a source term for the open water could be added here (as in Stern and Rothrock, 1995) to represent sub-grid scale sub-grid-scale convergence/divergence.

(b) Compute the new thin ice concentration as:

$$A_{t}_{\underline{\ \ }}^{n+1'} = \max(0,\min(1,1-A-A_{0}))$$
(A20)

(c) Compute the transfer of thin ice if  $\frac{A_t^{n+1}}{A_t} < A_t < A_t$  by setting:

$$H_t \underline{\overset{n+1'}{-}} = H_t \underbrace{\frac{A_t^{n+1}}{A_t}}_{t} \frac{A_t'}{A_t}$$

[revised manuscript text omitted]

---

## Author Response (AR2)

We thank the referee for reading thoroughly and performing a second round of reviews of the paper. We have incorporated most of the suggestions made. We comment on the exceptions below.

**1- p.6, line 19: I think the reference for Hutter et al. (2019) should be Hutter and Losch (2019) instead:**
**Hutter, N. and Losch, M.: Feature-based comparison of sea-ice deformation in lead-resolving sea-ice simulations, The Cryosphere Discuss., https://doi.org/10.5194/tc-2019-88, in review, 2019.**

Thank you, we have corrected this reference.

**2- p.6, line 20: "They report inconsistent temporal scaling with a reasonably good temporal scaling"??? and later: "and no temporal scaling in the region covered by the EGPS data they compare to." There is a temporal scaling, however the scaling exponent is smaller than in observations. I would correct this sentence to:**
**"They also report a reasonably good temporal scaling, however the scaling exponents found for the model simulation in the same region covered by the EGPS data are smaller than in observations."**

We agree that the formulation used was indeed not clear and we therefore rephrase this sentence as:
"The results on temporal scaling show some inconsistencies: the authors report a reasonably good temporal scaling when considering the full domain (but do not report on multi-fractality), however, in the smaller region covered by the EGPS data the estimated scaling exponants are significantly smaller than for observations."

**3- p.6, line 24: "but as this paper is still under review further detailing of their results is premature." The results shown in Hutter and Losch (2019) (reference above) are pretty conclusive. Please remove and add instead: "Hutter and Losch (2019) present further results confirming the ability of viscous-plastic sea-ice models run at very high resolution to reproduce the observed spatial and temporal scaling and multi-fractal behaviour of the ice."**

This is a subjective comment: the "conclusive" character of these results, as in any other study, should be assessed by peer-reviewing. Hence the sentence remains as is.

**4- p.7, line 14: "atmosphere–ocean interaction" --> "atmosphere–ocean interactions"**

Thank your for catching this.

**5- p.9, line 23: "The provided fields surface height fields" --> "The provided surface height fields"**

And again.

**6- p. 10, line 2: "The final ocean currents forcing" --> "The final ocean current forcing"**

Ok.

**7- The methodology for the scaling analysis is much clearer! Thank you. However, to avoid confusion, I would change "polygon" to "triangle" everywhere in this section since you only are considering triplets of points, i.e. triangles.**

We will keep polygons since it is a generic term for the method used: indeed, similar analysis (I.e., using contour integrals) could be performed on different types of polygons (e.g., squares).

**8- p. 13, line 6: "A is the encompassed area of the polygon equal to L^2." --> "A is the encompassed area of the polygon approximately equal to L2." Not all triangle areas will be exactly equal to L^2 if L is a mean for all triangles.**

Correct. We have corrected the text as suggested.

**9- p.14, line 26: "This effect is even more important..." This was already mentioned above at line 23 in the sentence starting with "In the time domain,..." Please remove one or the other.**

This last sentence was indeed redundant and we have removed it. We have modified line 23 slightly to convey the idea of line 24 and following.

**10- p.15, line 4: "total, shear and absolute deformation rates" --> "total, shear and absolute divergence deformation rates"**

Yes, thank you.

**11- p.15, line 6: "all simulated or observed deformation rates for the period of 7 days" --> "all simulated or observed 3-day deformation rates for a period of 7 days"**

Ok.

**12- p.15, line 23: "However, the first, second moments" --> "However, the first and second moments"**

Ok.

**13- p.16, line 3: The reference to Bouchat and Tremblay (2017) should be inserted after point (2) instead of after point (3) in this sentence.**

No: our mechanical parameters are not the same as in Tremblay and Bouchat, who used a VP rheology. A correct reference here would be Weiss and Dansereau, 2017, who performed a sensitivity analysis of the MEB model to the value of its parameters.
Here, the Bouchat and Tremblay citation indeed refers to the drag coefficient.

**14- p.20, line 20: "We note that using a Lagrangian mesh then helps preserving such features, once formed, but plays no role in their formation." I still think it does matter in their formation as well. It is easier to resolve discontinuities with a Lagrangian mesh compared to an Eulerian one, and therefore it will be easier for those features to appear with a Lagrangian mesh. Please remove this sentence.**

This is not true: the constitutive and dynamical equations solved are the same in a Lagrangian and a Eulerian frame, except the advective terms, which do not create velocity or stress gradients. The equations are well-posed and the choice of numerical scheme is independent of the physics. A Lagrangian scheme helps "resolving" gradients, but does not create them. This sentence stays as we believe it is in fac very important to make the distinction between the equations (and physics behind) and the numerical scheme used to solve them, as it seems to be a source of confusion.

**15- p.21, line 26: "that is about to be submitted." --> "in preparation."**

Agreed.

**16- Figure 1: Why do you have to mask the model field? If what you are plotting are the triangles that correspond to the ones in RGPS, then you should show all of them even if they cover different regions since they are entering your analysis.**

We are not masking the model fields. This was an incorrect sentence that we forgot to remove from the caption. We have therefore removed it now.